# Natural Bioactive Thiazole-Based Peptides from Marine Resources: Structural and Pharmacological Aspects

**DOI:** 10.3390/md18060329

**Published:** 2020-06-24

**Authors:** Rajiv Dahiya, Sunita Dahiya, Neeraj Kumar Fuloria, Suresh Kumar, Rita Mourya, Suresh V. Chennupati, Satish Jankie, Hemendra Gautam, Sunil Singh, Sanjay Kumar Karan, Sandeep Maharaj, Shivkanya Fuloria, Jyoti Shrivastava, Alka Agarwal, Shamjeet Singh, Awadh Kishor, Gunjan Jadon, Ajay Sharma

**Affiliations:** 1School of Pharmacy, Faculty of Medical Sciences, The University of the West Indies, St. Augustine, Trinidad & Tobago; Satish.Jankie@sta.uwi.edu (S.J.); Sandeep.Maharaj@sta.uwi.edu (S.M.); Shamjeet.Singh@sta.uwi.edu (S.S.); 2Department of Pharmaceutical Sciences, School of Pharmacy, University of Puerto Rico, Medical Sciences Campus, San Juan, PR 00936, USA; 3Department of Pharmaceutical Chemistry, Faculty of Pharmacy, AIMST University, Semeling, Bedong 08100, Kedah, Malaysia; neerajkumar@aimst.edu.my (N.K.F.); shivkanya_fuloria@aimst.edu.my (S.F.); 4Institute of Pharmaceutical Sciences, Kurukshetra University, Kurukshetra 136119, Haryana, India; sureshmpharma@rediffmail.com; 5School of Pharmacy, College of Medicine and Health Sciences, University of Gondar, P.O. Box 196, Gondar 6200, Ethiopia; ritz_pharma@yahoo.co.in; 6Department of Pharmacy, College of Medical and Health Sciences, Wollega University, P.O. Box 395, Nekemte, Ethiopia; sureshchennupati@rediffmail.com; 7Arya College of Pharmacy, Dr. A.P.J. Abdul Kalam Technical University, Nawabganj, Bareilly 243407, Uttar Pardesh, India; drhemendragautam@gmail.com; 8Department of Pharmaceutical Chemistry, Ideal Institute of Pharmacy, Wada, Palghar 421303, Maharashtra, India; rssunil29@gmail.com; 9Department of Pharmaceutical Chemistry, Seemanta Institute of Pharmaceutical Sciences, Jharpokharia, Mayurbhanj 757086, Orissa, India; sanjay_karan21@rediffmail.com; 10Department of Pharmaceutical Chemistry, The Oxford College of Pharmacy, Hongasandra, Bangalore 560068, Karnataka, India; jyotishrivastavapharmacy@gmail.com; 11Department of Pharmaceutical Chemistry, U.S. Ostwal Institute of Pharmacy, Mangalwad, Chittorgarh 313603, Rajasthan, India; agarwalalka2014@gmail.com; 12Department of Pharmaceutical Biotechnology, Shrinathji Institute of Pharmacy, Nathdwara 313301, Rajsamand, Rajasthan, India; awadh.k1771@gmail.com; 13Department of Pharmaceutical Chemistry, Shrinathji Institute of Pharmacy, Nathdwara 313301, Rajsamand, Rajasthan, India; jadon_gunjan@yahoo.in; 14Department of Pharmacognosy and Phytochemistry, School of Pharmaceutical Sciences, Delhi Pharmaceutical Sciences and Research University, New Delhi 110017, India; ajaysharmapharma1979@gmail.com

**Keywords:** azole-based peptide, marine sponge, peptide synthesis, cytotoxicity, cyanobacteria, thiazole, bioactivity

## Abstract

Peptides are distinctive biomacromolecules that demonstrate potential cytotoxicity and diversified bioactivities against a variety of microorganisms including bacteria, mycobacteria, and fungi via their unique mechanisms of action. Among broad-ranging pharmacologically active peptides, natural marine-originated thiazole-based oligopeptides possess peculiar structural features along with a wide spectrum of exceptional and potent bioproperties. Because of their complex nature and size divergence, thiazole-based peptides (TBPs) bestow a pivotal chemical platform in drug discovery processes to generate competent scaffolds for regulating allosteric binding sites and peptide–peptide interactions. The present study dissertates on the natural reservoirs and exclusive structural components of marine-originated TBPs, with a special focus on their most pertinent pharmacological profiles, which may impart vital resources for the development of novel peptide-based therapeutic agents.

## 1. Introduction

Heterocycles are known to govern a lot of processes of vital significance inside our body, including transmission of nerve impulses, hereditary information, and metabolism. A variety of the naturally occurring congeners, including reserpine, morphine, papaverine, and quinine, are heterocycles in origin, and many of the synthetic bioactives viz. methotrexate and isoniazid contain heterocyclic pharmacophores [1]. Among heterocycles, thiazoles have received special attention as promising scaffolds in the area of medicinal chemistry because this azole has been found alone or incorporated into the diversity of therapeutic active agents such as sulfathiazole, combendazole, niridazole, fanetinol, bleomycin, and ritonavir, which are associated with antibiotic, fungicidal, schistozomicidal, anti-inflammatory, anticancer, and anti-HIV properties [2,3]. Peptides are bioactive compounds of natural origin available in all living organisms and are known for their vital contribution in a wide array of biological activity. Due to their therapeutic abilities, peptides have received growing interest in recent years. In the human body, peptides perform a lot of essential functions including the engagement of peptide hormones like insulin, glucagon-like peptide-1 (GLP-1), and glucagon and in blood glucose regulation and are used to treat novel targets for certain disease conditions, including Alzheimer’s disease, diabetes mellitus type 2, and obesity [4,5,6,7].

As unique structural features make azole-containing heterocyclic peptides (especially thiazoles) attractive lead compounds for drug development as well as nice tools for advance research, efforts should be made by scientists to develop biologically active thiazole-based peptide derivatives (TBPs). TBPs are obtained from diverse resources, primarily from cyanobacteria, sponges, and tunicates. A thiazole ring can be part of a cyclic structure or connected in a linear chain of peptides either alone or with other heterocycles like oxazole (e.g., thiopeptide antibiotics), imidazole, and indole (in the forms of histidine and tryptophan), thiazoline, oxazoline, etc. Cyclic peptides have an advantage over their linear counterparts as cyclization offers a reduction in conformational freedom, resulting in higher receptor-binding affinities. Understanding the structure–activity relationship (SAR), different modes of action, and routes of synthesis as tools are of vital significance for the study of complex molecules like heterocyclic bioactive peptides, which have a broad spectrum of pharmacological activities associated with them. Further, the sudden increase in the number of peptide drug products is another good reason to study this particular category of compounds on a priority basis. Keeping in view the vital significance of TBPs, the current article focuses on different bioactive marine-derived thiazole-based polypeptides with complex structures and their potent resources, synthetic methodologies, stereochemical aspects, structural activity relationships, diverse modes of action, and bioproperties.

### 1.1. Resources

Various natural sources of TBPs and other heterocyclic rings containing cyclopolypeptides comprise cyanobacteria [8,9,10,11,12,13,14,15,16,17,18,19,20,21,22,23,24,25,26,27,28,29,30,31,32,33,34,35,36,37,38,39,40], ascidians [41,42,43,44,45,46,47,48,49,50,51,52,53,54,55,56,57,58,59,60,61,62], marine sponges [63,64,65,66,67,68,69,70], and sea slugs [71,72,73]. Moreover, actinomycetes, sea hare, red alga, and higher plants [74,75,76,77,78,79,80] were found to be other potential resources of TBPs.

### 1.2. Linear vs. Cyclic Peptides

In linear peptides with amino acid units between 10 to 20, secondary structures like α-helices and β-strands begin to form, which impose constraints that reduce the free energy of linear peptides. Compared to linear peptides, cyclopeptides are typically considered to have even greater potential as therapeutic agents due to their increased chemical and enzymatic stability, receptor selectivity, and improved pharmacodynamic properties. Although peptide cyclization generally induces structural constraints, the site of cyclization within the sequence can affect the binding affinity of cyclic peptides. Cyclization is a well-known technique to increase the potency and in vivo half-life of peptide molecules by locking their conformation. Hence, both the biological activity and the stability of peptides can be improved by cyclization. The reduction in conformational freedom brought about by cyclization often results in higher receptor-binding affinities. Overall, cyclization of peptides is a vital tool for structure–activity studies and drug development because ring formation limits the flexibility of the peptide chain and allows for the induction or stabilization of active conformations. Moreover, cyclic peptides are less sensitive to enzymatic degradation [81].

The cyclization process often increases the stability of peptides, can prolong their bioeffect, and can create peptides with the ability to penetrate tumors in order to enhance the potency of anticancer drugs [82,83]. Cyclization is envisioned to enhance the selective binding, uptake, potency, and stability of linear precursors. The prolonged activity may even be the result of additional resistance to enzymatic degradation by exoproteases. Cyclic peptides are of considerable interest as potential protein ligands and might be more cell permeable than their linear counterparts due to their reduced conformational flexibility.

Further, cyclic nature of peptides was found to be crucial to their bioactivity in the case of depsipeptides. For example, corticiamide A is a member of a family of structurally related cyclic depsipeptides with tryptophan moiety that include the discodermins, halicylindramides, polydiscamide A, and microspinosamide A. However, corticiamide A is the only member of the family to contain a *p*-Br-Phe at residue 11 and an *N*-MeAsn. Microspinosamide A and polydiscamide A contained the unusual β-Me-Ile at residue 6, whereas the same amino acid is found at residue 5 in corticiamide A. All these peptides were known to be cytotoxic in the low µM range and to inhibit the growth of bacteria and fungi in addition to inhibition of the cytopathic effect of HIV-1 in mosaic human T cell leukemia cells-Syncitial Sensitive (CEM-SS) by microspinosamide A. Interestingly, the cyclic nature of these peptides was important for their bioactivity, with linear versions exhibiting a loss of activity of at least 1 order of magnitude [84,85].

## 2. Chemistry

### 2.1. Structural Features of Thiazole (Tzl)-Containing Cyclooligopeptides

Aestuaramides, banyascyclamides, ulongamides (**1**–**3**), guineamides (**4**,**5**), microcyclamides MZ602 and MZ568, trichamide, tawicyclamides (**6**,**7**), obyanamide (**8**), cyclodidemnamide and cyclodidemnamide B, lyngbyabellins, oriamide (**9**), scleritodermin A (**10**), haligramide A (**11**), waiakeamide (**12**), haligramide B (**13**), mollamide C (**15**), jamaicensamide A (**16**), myotamides, didmolamides, dolastatin 3, homodolastatin 3, sanguinamides, cyclotheonellazoles, aeruginazole A, aeruginazole DA1497, aeruginazole DA1304, and aeruginazole DA1274 are examples of heterocyclic thiazole-based polypeptides having diverse unusual structural features from marine organisms.

Cyanobactin cyclopolypeptide aestuaramide A contained valylthiazole (Val-Tzl) and prolylthiazole (Pro-Tzl) residues in addition to proline, valine, and methionine units and a reverse *O*-Tyr isoprene moiety (Ptyr). Aestuaramide B was found to be an unprenylated analogue of aestuaramide A, whereas aestuaramide C was found to be a forward C-prenylated derivative. Aestuaramide D–F and aestuaramide J–L were found to be the sulfoxide derivatives of aestuaramides A–C and aestuaramides G–I, respectively. Similarily, aestuaramides G−L were reverse O-prenylated, unprenylated, or forward C-prenylated congeners, with or without Met oxidation, but contained alanylthiazole (Ala-Tzl) instead of a Val-Tzl unit of aestuaramides A–F. Cyclic peptides such as aestuaramides may be exceptionally widespread metabolites in natural ecosystems [10].

Banyascyclamides B and C are modified cyclopolypeptides, closely related in structure, and composed of two thiazole-alanine units. The cyclohexapeptide banyascyclamide C exhibited close structural similarity with banyascyclamide A but differed in having l-phenylalanyl-l-threonine moiety instead of l-Phe-mOzl residue of banyascyclamide A. Similarily, banyascyclamide B differed from banyascyclamide C in having l-leucyl-l-threonine moiety instead of l-phenylalanyl-l-threonine residue [11].

The cyanobacterium-derived ulongamide A (**1**) and other ulongamides B–F are alanine-derived thiazole carboxylic acid (l-Ala-Tzl-ca) containing cyclodepsipeptides which possessed a novel β-amino acid residue, 3-amino-2-methylhexanoic acid (Amha). Further, there was the presence of 2-hydroxyisovaleric acid (Hiva) in ulongamide D (**2**) and 2-hydroxy-3-methylpentanoic acid (Hmpa) in ulongamide E and ulongamide F (**3**), which had replaced the l-lactic acid moiety present in ulongamides A–C. Ulongamides A–E displayed weak in vitro cytotoxicity against ubiquitous KERATIN-forming tumor cell subline (KB) and LoVo cells [13] (Figure 1).

The cyanobacterium-derived guineamide A (**4**) contained the common l-alanine-disubstituted- thiazole unit, unique β-amino acid 2-methyl-3-aminopentanoic acid (Mapa), lactic acid (l-Lac), *N*-methylated amino acids viz. *N*-methylphenylalanine (l-*N*-MePhe), and *N*-methylvaline (l-*N*-MeVal), but guineamide B (**5**) deviated from guineamide A (**4**) in having 2-hydroxyisovaleric acid (l-Hiv) and 2-methyl-3-aminobutanoic acid (Maba) units instead of l-Lac and Mapa units. The absolute stereochemistry of the 2-methyl-3-aminopentanoic acid (Mapa) unit in guineamide A (**4**) was found to be 2*S*,3*R*. From a biosynthetic perspective, the guineamides were found to be interesting molecules because of the presence of unusual α-amino and β-hydroxy acid residues. Further, guineamide B (**5**) exhibited moderate cytotoxic activity against a mouse neuroblastoma cell line [14] (Figure 2).

Microcyclamides MZ602 and MZ568 contained isoleucylthiazole moiety in common but differed in having phenylalanine and glycine amino acids in the former and valine and alanine in the latter. Trichamide possessed serylthiazole and leucylthiazole moieties in addition to histidine amino acid [18].

The cyanobacterium-derived lyngbyabellin A is a significantly cytotoxic dichlorinated peptolide with unusual structural features, including a dichlorinated α-hydroxy acid and two functionalized thiazole carboxylic acid units. This depsipeptide was found to be a potent disrupter of the cellular microfilament network [27]. Lyngbyabellin B is related cyclic depsipeptide in which one thiazole unit was replaced by a thiazoline ring, with the placement of the ring between the glycine residue and the α,β-dihydroxyisovaleric acid rather than adjacent to the valine-derived unit, and the isoleucine-derived unit in lyngbyabellin A was replaced by a valine-derived moiety in lyngbyabellin B. Lyngbyabellin B displayed potent toxicity toward brine shrimp and the fungus *Candida albicans* and was found to be slightly less cytotoxic in vitro than lyngbyabellin A against KB and LoVo cells, respectively [86]. The structures of lyngbyabellin E and H showed the presence of two 2,4-disubstituted thiazole rings and differed7 in having the α,β-dihydroxyisovaleric acid (dhiv) unit in lyngbyabellin E replaced by the 2-hydroxyisovaleric acid (hiva) unit in lyngbyabellin H. Intriguingly, lyngbyabellin E and H appeared to be more active against the H460 human lung tumor cell lines. From the bioactivity results, it appeared that lung tumor cell toxicity is enhanced in the cyclic representatives with an elaborated side chain [28].

In addition to two thiazole rings and a chlorinated 2-methyloctanoate residue, lyngbyabellin N contained an unusual dimethylated valine terminus and a leucine statine residue. The planar structure of lyngbyabellin N was closely related to that of lyngbyabellin H except for the replacement of the polyketide portion with an *N,N*-dimethylvaline (DiMeVal) residue [29]. The cytotoxic lyngbyabellin J contained the *gem*-dichloro moiety as part of a 7,7- dichloro-3-acyloxy-2-methyloctanoate residue in addition to the α,β-dihydroxy-β-methylpentanoic acid (Dhmpa, C_19–24_) unit and two disubstituted thiazole rings [30].

Tawicyclamides A and B (**6**,**7**) represent a novel category of cyclooligopeptides, bearing alternative sequences of two thiazoles and one thiazoline amino acid but lacking the oxazoline ring, which is characteristic of ascidian-derived heptapeptides lissoclinamides and the octapeptides patellamides/ulithiacyclamides. Moreover, the presence of a *cis*-valine-proline amide bond facilitates an unusual three-dimensional conformation to ascidian-derived tawicyclamides A and B (**6**,**7**). Tawicyclamide B (**7**) differs from tawicyclamide A (**6**) in having a leucine moiety in place of the phenylalanine residue of tawicyclamide A [41] (Figure 3).

In the structure of depsipeptide–obyanamide (**8**), the alanylthiazole (Ala-Tzl) unit and 3-aminopentanoic acid (Apa) were present [12,42] whereas the sponge-derived cytotoxic cyclic peptide, oriamide (**9**), was found to contain a new 4-propenoyl-2-tyrosylthiazole amino acid (PTT) moiety. Further, a novel conjugated thiazole moiety viz. 2-(1-amino-2-*p*-hydroxyphenylethane)-4- (4-carboxy-2,4-di-methyl-2*Z*,4*E*-propadiene)-thiazole (ACT) was found to be part of the structure of tubulin inhibitory sponge-derived cyclopolypeptide scleritodermin A (**10**), along with *O*-methyl-*N*-sulfoserine and keto-*allo*-isoleucine units [64] (Figure 4).

In the structure of the bisthiazole-containing macrocyclic peptide, cyclodidemnamide B, two thiazole moieties viz. prolylthiazole (l-Pro-Tzl) and leucylthiazole (d-Leu-Tzl) were found to be present. The ascidian-derived cyclodidemnamide was found to be similar to reverse prenyl substituted cytotoxic cycloheptapeptide mollamide only in possessing the same dihyrothiazole-proline dipeptide unit (C_20_–C_27_), but it also contained leucylthiazole and phenylalanyl-methyl oxazoline moieties [43,62].

The sponge-derived cytotoxic hexapeptides haligramide A and B (**11**,**13**) were found to contain the phenylalanylthiazole (Phe-Tzl) moiety in addition to three proline units. Haligramide A (**11**) was the bismethionine analogue of waiakeamide (**12**), bearing Phe-Tzl moiety. Haligramide B (**13**) contained both methionine and methionine sulfoxide residues in comparison to haligramide A (**11**) which contained only methionine residues and waiakeamide (**12**), another sponge-derived cyclohexapeptide that contained methionine sulfoxide residues only [63,66] (Figure 5).

A unique amino acid, 2-bromo-5-hydroxytryptophan (BhTrp), and an unusual ureido linkage were found to be present in the composition of sponge-derived peptide konbamide with calmodulin antagonistic activity [87]. Further, the cytotoxic depsipeptide polydiscamide A contained a novel amino acid 3-methylisoleucine in addition to heterocyclic tryptophan moiety [65,88].

The notaspidean mollusk-derived cytotoxic cyclic hexapeptide keenamide A (**14**) contained a leuylthiazoline (Leu-Tzn) unit together with serylisoprene residue in its structure and differed from mollamide C (**15**), a tunicate-derived cyclohexapeptide, in having thiazoline moiety instead of thiazole [72]. Trunkamide A contained a thiazoline heterocycle and two residues of Ser and Thr with the hydroxy function modified as reverse prenyl (rPr). The structure of jamaicensamide A (**16**), a sponge-derived peptide having β-amino-α-keto and thiazole-homologated η-amino acid residues, was found to contain 2-aminobutanoic acid (Aba), 5-hydroxytryptophan (HTrp), and a terminal 2-hydroxy-3-methylpentanamide (Hmp) unit [44,89] (Figure 6).

Myotamides A and B are ascidian-derived cycloheptapeptides that contained three unusual amino acids containing heteroatoms including one thiazole (Tzl) and two thiazoline (Tzn) rings in addition to valine, proline, isoleucine, and methionine. Mayotamide A embodied the same Val-Pro-Tzn sequence as was found in ascidian-derived cyclic heptapeptide cyclodidenmamide and also contained an additional thiazoline (Tzn) ring. Myotamide A differed from myotamide B in having isoleucine moiety, which was replaced by valine moiety in the latter. Both cyclopolypeptides exhibited cytotoxicity against tumor cell lines [45].

Didmolamide B is a thiazole-containing ascidian-derived cyclopolypeptide that contained two l-alanylthiazole residues, and l-phenylalanine and l-threonine moieties. The threonine residue of didmolamide B was modified to a methyloxazoline (mOzn) heterocycle in the case of didmolamide A. Didmolamide B was found to exhibit mild cytotoxicity against several cultured tumor cell lines [48].

Dolastatin 3 is a cyanobacterium- as well as sea hare-derived cyclopolypeptide that contained two l-glutaminyl-thiazole (l-Gln-Tzl) and glycyl-thiazole (Gly-Tzl) units in addition to l-valine, l-leucine, and l-proline residues. The cyanobacterium-derived homodolastatin 3 differed from dolastatin 3 by the addition of a methylene group, i.e., an l-isoleucine residue in place of the l-valine residue of dolastatin 3. The cyclopentapeptide dolastatin 3 was found to exhibit HIV-1 integrase inhibitory activity as well as P388 lymphocytic leukemia (PS) cell growth inhibitory activity. Kororamide is another cyanobacterium-derived polypeptide having two l-tyrosinyl-thiazole (l-Tyr-Tzl) and leucyl-thiazoline (Leu-Tzn) units in addition to l-leucine, l-isoleucine, l-serine, l-proline, and l-asparagine residues [9,90].

The sponge-derived cyclotheonellazoles A–C are unusual cyclopolypeptides containing nonproteinogenic acids, the most unique being 4-propenoyl-2-tyrosylthiazole (PTT), 3-amino-4-methyl-2-oxohexanoic acid (Amoha), and diaminopropionic acid (Dpr), along with two or three proteinogenic amino acids like glycine and alanine. Cyclotheonellazoles B and C shared the same basic structure with cyclotheonellazole A, in which leucine (in cyclotheonellazole B) and homoalanine (in cyclotheonellazole C) replaced the 2-aminopentanoic acid residue of cyclotheonellazole A. Cyclotheonellazoles were found to be nanomolar inhibitors of chymotrypsin and sub-nanomolar inhibitors of elastase [68].

The nudibranch-derived sanguinamide A is a modified heptapeptide containing a 2-substituted thiazole-4-carboxamide moiety. Structural analysis of this peptide indicated the presence of two residues, l-proline and l-isoleucine, present in alternative continuous sequences in addition to amino acid moieties phenylalanine and alanine with an l-configuration. In this cycloheptapeptide, azole-modified amino acid was found to be l-isoleucyl-thiazole (l-Ile-Tzl). In comparison to sanguinamide A, the cyclic octapeptide sanguinamide B was found to contain additional heteroaromatic oxazole and thiazole rings [73].

The cyanobacterium-derived polythiazole peptide aeruginazole DA1497 contained leuylthiazole (Leu-Tzl), alanylthiazole (Ala-Tzl), phenylalanylthiazole (Phe-Tzl), and valylthiazole (Val-Tzl) residues and exhibited bioproperties against Gram-positive bacterium *Staphylococcus aureus*. However, in related cyclopolypeptides, aeruginazole DA1304 and aeruginazole DA1274 moieties like asparaginylthiazole (Asn-Tzl), Leu-Tzl and isoleucylthiazole (Ile-Tzl) were found to be present. l-Asn-Tzl moiety was also observed in the polythiazole containing cyanobacterium-derived polypeptide aeruginazole A in addition to d-Leu-Tzl and l-Val-Tzl residues. This cyclododecapeptide was found to potently inhibit the Gram-positive bacterium *Bacillus subtilis* [8,91].

### 2.2. Structural Features of Tzl-Containing Linear Peptides

In addition to cyclopolypeptides, heterocyclic thiazole ring-based linear peptides are also obtained from marine organisms. Micromide (**17**), apramides (**18**,**19**), dolastatin 10 (**20**), symplostatin 1 (**21**), dolastatin 18 (**22**), lyngbyapeptins A and C (**23**,**24**), and lyngbyabellin F (**25**) and I (**26**) are the best examples of linear peptides containing thiazole rings.

Micromide (**17**) is a highly *N*-methylated linear peptide containing structural features common to many cyanobacterial metabolites, including a d-amino acid, a modified cysteine unit in the form of a thiazole ring and *N*-methylated amino acids. The structrural components of this peptide included moieties like 3-methoxyhexanoic acid, *N*-Me-Gly-thiazole, and other *N*-methylated amino acids viz. *N*-Me-Phe, *N*-Me-Ile, *N*-Me-Val, etc. Micromide (**17**) was found to exhibit cytotoxicity against KB cells [92]. On the other hand, the cyanobacterium-derived apramides A–G are linear lipopeptides containing a thiazole-containing modified amino acid unit. Structural analysis of apramide A (**18**) suggested the presence of a 2-methyl-7-octynoic acid moiety (Moya) and six amino acid residues (*N*-Me-Ala, Pro, *N*,*O*-diMe-Tyr, and 3 units of *N*-Me-Val) and a C-terminally modified amino acid unit (*N*-Me-Gly-thz). Structures of apramide B and apramide C (**19**) differed from apramide A (**18**) in having the presence of a 7-octynoic acid unit (Oya) and 2-methyl-7-octenoic acid moiety (Moea) in lieu of the Moya moiety of apramide A (**18**). Apramides D–F differed from apramide A (**18**), B, and C (**19**), only by bearing a Pro-Tzl unit instead of the *N*-Me-Gly-Tzl residue, which had caused a drastic impact on the conformational behavior. The lipopeptide apramide A (**18**) was found to enhance elastase activity [93] (Figure 7).

The dolastatins are sea hare- and marine cyanobacterium-derived compounds that exhibit cytotoxic properties. Dolastatin 10 (**20**) is a linear thiazole-containing heterocyclic peptide bearing *N*,*N*-dimethylvaline, (3*R*,4*S*,5*S*)-dolaisoleucine, (2*R*,3*R*,4*S*)-dolaproine, and (*S*)-dolaphenine [94]. Like dolastatin 10 (**20**), cyanobacterium-derived symplostatin 1 (**21**) is a potent microtubule inhibitor. Symplostatin 1 (**21**) differed from dolastatin 10 (**20**) by the replacement of the *iso*-propyl group by a *sec*-butyl group on the first *N*-dimethylated amino acid. Symplostatin 1 (**21**) is a very potent cytotoxin but not as potent as dolastatin 10 (**20**), whereas synthetic analogues lacking the *N,N*-dimethylamino acid residue were reported to be markedly less cytotoxic. The structure of symplostatin 1 (**21**) differed from dolastatin 10 (**20**) by only one additional CH_2_ unit in the *N*-terminal residue. The absolute configuration of the stereocenter at C-26 in symplostatin 1 (**21**) was found to be 26*S*. The biological evaluation of symplostatin 1 (**21**) revealed that it is highly active against certain tumors and comparable in its activity with isodolastatin H. Both dolastain 10 (**20**) as well as its methyl analog, symplostatin 1 (**21**) were found to be potent microtubule depolymerizers [95,96].

Dolastatin 18 (**22**) is another cancer cell growth inhibitory linear peptide bearing thiazole moiety from the sea hare, the structure of which is derived from two α-amino acids (Leu and MePhe), a dolaphenine (Doe) unit, and the new carboxylic acid 2,2-dimethyl-3-oxohexanoic acid (dolahexanoic acid, Dhex). Dolastatin 18 (**22**) was found to significantly inhibit growth of human cancer cell lines [97] (Figure 8).

Lyngbyapeptins are thiazole-containing lipopeptides with a rare 3-methoxy-2-butenoyl moiety with a high level of *N*-methylation. The cyanobacterium-derived lyngbyapeptin A (**23**) is a linear modified peptide with a 2-substituted thiazole ring. In comparison to lyngbyapeptin A (**23**), lyngbyapeptin B and C possess the same/similar characteristic C- and N-terminal modification and differed by containing other amino acid units in between. Structural analysis of lyngbyapeptin B indicated the presence of two *N*,*O*-dimethyltyrosine residues, an *N*-methylvaline unit, a thiazole-containing modified alanine (Ala-thz) unit, and a 3-methoxy-2-butenoic acid (Mba) moiety with the absolute stereochemistry *S* for the methylated amino acids. The structure of lyngbyapeptin C (**24**) differed from that of lyngbyapeptin B in having the presence of an *N*-terminal unit and 3-methoxy-2-pentenoic acid (Mpa) residue. The structure of lyngbyapeptin D (**27**) differed from that of lyngbyapeptin A (**23**) in having *N*-Me-Val residue instead of *N*-Me-Ile in addition to *N*-Me-Leu, a thiazole-containing modified proline (Pro-thz) unit and *N*,*O*-dimethyltyrosine (*N*,*O*-diMe-Tyr) [98,99]. Lyngbyabellin F (**25**) and I (**26**) are linear dichlorinated lipopeptides that showed the presence of two 2,4-disubstituted thiazole rings. Lyngbyabellin I (**26**) and F (**25**) were found to be cytotoxic to human lung tumor and neuro-2a mouse neuroblastoma cells [100] (Figure 9).

### 2.3. Structural Features of Thiazole (Tzl)- and Oxazole (Ozl)-Containing Cyclopeptides

In addition to cyclic peptides with thiazole/thiazoline rings, mixed heterocyclic ring-based cyclopeptides are also derived from marine resources. Comoramide A, didmolamides A–C (**28**–**30**), vemturamides (**31**,**32**), dolastatins E and I (**34**,**35**), microcyclamide (**36**), bistratamides (**37**–**41**), raocyclamides (**42**,**43**), tenuecyclamides, patellamides, and lissoclinamides are bioactive cyclooligopeptides containing thiazole and oxazole rings.

Comoramides are cyanobactins that contained prenylated amino acids. The ascidian-derived cyclopeptide comoramide A was isolated with threonine heterocyclized in position 5 and prenylated in position 3 and was found to contain six amino acids in its structure, including two amino acids that existed as a 5-methyloxazoline (mOzn) heterocycle and as a thiazoline ring (Tzn). The additional amino acid moieties present were l-alanine, l-phenylalanine, and l-isoleucine. Like patellin, trunkamide A, mollamide, and hexamollamide, comoramide A was found to be a unique type of peptide that contained threonine residue for which the side chain is modified as dimethylallyl ether. This cyclohexapeptide exhibited structural similarilty with another ascidian-derived cycloheptapeptide mollamide in two amino acids viz. Ile-Tzn and Phe-Thr. Comoramide A was found to be cytotoxic against the A549, HT29, and MEL-28 tumor cell lines [45].

Didmolamides A and B (**28**,**29**) are ascidian-derived cyclohexapeptides that contained two l-alanylthiazole residues and one l-phenylalanine moiety in common but didmolamide A (**28**) contained 5-methyloxazoline (mOzn) heterocycle in addition, which is replaced by l-threonine moiety in didmolamide B (**29**). Morover, didmolamide C (**30**) differs from didmolamides A and B (**28**,**29**) in the oxidation state of the heterocyclic rings, having two thiazoline rings (instead of thiazoles) in didmolamide C (**30**). Additionally, didmolamide C (**30**) was found to contain a methyloxazole ring instead of a methyloxazoline ring of didmolamide A (**28**). Didmolamide A (**28**) displayed mild cytotoxicity against the A549, HT29, and MEL28 tumor cell lines [48,101] (Figure 10).

Venturamides (**31**,**32**) are cyanobacterium-derived thiazole- and methyloxazole-containing cyclohexapeptides that exhibited antimalarial and cytotoxic activities. Structural analysis of venturamide B (**32**) indicated the presence of d-alanine, d-valine, and d-*allo*-threonine in addition to three heteroaromatic moieties. The polypeptide venturamide B (**32**) was identified as cyclo-d-*allo*-Thr-Tzl-d-Val-Tzl-d-Ala-mOzl. The cyclic hexapeptide venturamide B (**32**) differed from venturamide A (**31**) in having a d-threonine unit in place of the d-alanine adjacent to the thiazole ring. There was a close similarity between the structures of venturamide A (**31**) and blue-green alga-derived cyclopeptide dendroamide A (**33**): however, d-valine and d-alanine are exchanged with each other, adjacent to two thizaole heterocycles at C-12 and C-20. Venturamides (**31**,**32**) showed strong in vitro activity against *Plasmodium falciparum*, with only mild cytotoxicity to mammalian Vero cells. Also, mild activity against *Trypanasoma cruzi*, *Leishmania donovani*, and MCF-7 cancer cells was also reported for venturamides [34] (Figure 11). 

The sea hare-derived cyclopolypeptides dolastatins E and I (**34**,**35**) were found to contain three kinds of five-membered heterocycles viz. oxazole/methyloxazole (Ozl/mOzl), thiazole (Tzl), and thiazoline/oxazoline (Tzn/Ozn), in addition to one residue each of d-alanine and l-alanine and one residue of d-isoleucine in dolastatin E (**34**) while one residue each of l-alanine, l-valine, and l-isoleucine in the case of dolastatin I (**35**). Although both of these cyclic hexapeptides displayed cytotoxicity against HeLa S_3_ cells, in comparison, dolastatin I (**35**) was found to be more cytotoxic than dolastatin E [75,76]. On the other hand, in addition to two thiazole (Tzl) and one methyloxazole (mOzl) rings, the cyanobacterium-derived cyclopeptide microcyclamide (**36**) contained two usual amino acids, l-isoleucine and l-alanine, and one *N*-methylhistidine residue. Overall, the hexapeptidic structure was composed of three units viz. thiazole-methylhistidinyl, thiazole-isoleucinyl, and methyloxazole-alanyl units. This cyclic hexapeptide displayed a moderate cytotoxic activity against P388 murine leukemia cells [35] (Figure 12).

The ascidian-derived bistratamide A and B contained heteroaromatic rings viz. methyloxazoline (mOzn) and thiazoline (Tzn) rings in common in addition to one residue each of alanine, phenylalanine, and l-valine. However, bistratamide A differed from bistratamide B only in the conversion of one thiazoline ring to a thiazole, i.e., these hexapeptides differed only by the the presence or absence of one double bond. Both these cyclohexapeptides displayed activity toward human cell lines viz. MRC5CV1 fibroblasts and T24 bladder carcinoma cells. Bistratamides C and D (**37**,**38**) possessed one thiazole ring in common in addition to two l-valine residues. However, bistratamide C (**37**) differed from bistratamide D (**38**) in having an l-alanine moiety instead of additional l-valine. Moreover, the other two heteroaromatic rings in bistratamide D (**38**) were methyloxazoline and oxazole, whereas in bistratamide C (**37**), oxazole and thiazole rings were present. Bistratamides E and F were found to contain three residues of l-valine in addition to thiazole and methyloxazoline rings. Bistratamide F differed from bistratamide E in having an additional oxazoline ring instead of a second thiazole ring in bistratamide E. Similarily, bistratamides G and H (**39**,**40**) were found to contain three residues of l-valine in addition to thiazole and methyloxazole rings. Bistratamide G (**39**) differed from bistratamide H (**40**) in having an additional oxazole ring instead of a second thiazole ring in bistratamide H (**40**). Further, bistratamide I (**41**) contained three residues of l-valine in addition to one thiazole and one oxazole ring. The ascidian-derived bistratamides M and N (**46**,**47**) are oxazole-thiazole-containing cyclic hexapeptides that displayed moderate cytotoxicity against four human tumor cell lines including NSLC A-549 human lung carcinoma cells, MDA-MB-231 human breast adenocarcinoma cells, HT-29 human colorectal carcinoma cells, and PSN1 human pancreatic carcinoma cells. Moreover, bistratamides G-I (**39**–**41**) and J showed weak to moderate activity against the HCT-116 human colon tumor cell line [50,59,60,61] (Figure 13).

Raocyclamides (**42**,**43**) are cyclooligopeptides in which the ring system contains amide links only, and they contain three heteroaromatic rings symmetrically arranged in a peptide chain with different connected aliphatic amino acids providing structural diversity. Raocyclamides A and B (**42**,**43**) are cyanobacterium-derived oxazole- and thiazole-containing cyclic hexapeptides with cytotoxic properties. Raocyclamide A (**42**) contained three standard amino acid residues viz. d-isoleucine, l-alanine, and d-phenylalanine and three modified amino acids viz. thiazole, oxazole, and oxazoline. In comparison, raocyclamide B (**43**) contained four standard amino acid residues viz. d-isoleucine, l-alanine, d-phenylalanine, and d-serine and two modified amino acids viz. thiazole and oxazole. Raocyclamide A (**42**) differed from raocyclamide B (**43**) in having an additional heterocyclic ring “oxazoline” with a d-configuration instead of a d-serine residue. Raocyclamide A (**42**) was found to be moderately cytotoxic against sea urchin embryos [32] (Figure 14).

The ascidian-derived lissoclinamides 1–10 and cyanobacterium-derived tenuecyclamide A and B are other cyclopolypeptides containing thiazole, thiazoline, methyloxazole, and methyloxazoline rings which displayed cytotoxicity against SV40 transformed fibroblasts and transitional bladder carcinoma cells as well as inhibited the division of sea urchin embryos [102,103,104,105].

Various heterocyclic marine-derived thiazole-based cyclopolypeptides including those having thiazoline (Tzn), oxazole (Ozl), oxazoline (Ozn), 5-methyloxazole (mOzl), 5-methyloxazoline (mOzn), 5-hydroxytryptophan (Htrp), N-methylimidazole (mImz), histidine (His), tryptophan (Trp), 2-bromo-5-hydroxytryptophan (Bhtrp), and N-methyltryptophan (Metrp) rings in addition to thiazole, together with their molecular formulas and composition, are tabulated in Table 1.

### 2.4. Structural Features of Thiopeptide Antibiotics

Thiopeptides are a novel family of antibiotics which are associated with a lot of pharmacological properties including immunosuppressive, antineoplastic, antimalarial, and potent antimicrobial activity against Gram-positive bacteria. Due to their interesting structures and bioprofile against bacteria, thiopeptides have attracted the attention of researchers and scientists as a new class of emerging antibiotics. The most important characteristic feature of the thiopeptides is the central nitrogen-containing six-membered ring with diverse oxidation states. On the basis of different oxidation states of the central ring of thiopeptides, they can belong to the “a series” with a totally reduced central piperidine, the “b series” with a 1,2-dehydropiperidine ring, and the “c series” with a piperidine ring fused with imidazoline. All members of series a, b, and c have a macrocycle which contains a quinaldic acid moiety. The d series shows a trisubstituted pyridine ring, and the e series is known for the hydroxyl group in the central tetrasubstituted pyridine ring. The e series also presents a macrocycle formed by a modified 3,4-dimethylindolic acid moiety. The central ring in thiopeptides serves as a scaffold to at least one macrocycle and a tail, containing different thiazoles and oxazoles which are developed by dehydration/dehydrosulfanylation of amino acid like serine, cysteine, etc. TP-1161, YM-266183, YM-266184, kocurin, baringolin, geninthiocin, Ala-geninthiocin, and Val-geninthiocin are examples of thiopeptides from marine resources [111].

TP-1161 belongs to the “d series” of thiopeptide antibiotics, produced by a marine sediment-derived *Nocardiopsis* sp. Structural features of this thiopeptide include the three 2,4-disubstituted thiazoles and one 2,4-disubstituted oxazole moiety in addition to the presence of a trisubstitued pyridine (Pyr) functional unit and an unusual aminoacetone moiety. TP-1161 displayed good activity against a panel of Gram-positive bacteria including *Staphylococcus aureus*, *Staphylococcus haemolyticus*, *Staphylococcus epidermidis*, *Enterococcus faecium*, and *Enterococcus faecalis* [112].

YM-266183 and YM-266184 are novel thiopeptide antibiotics produced by *Bacillus cereus* isolated from a marine sponge and structurally related to a known family of antibiotics that include thiocillins and micrococcins. Structural analysis of these thiopeptides indicated the presence of several unusual amino acids with heteroaromatic moieties, including the six thiazole rings, a 2,3,6-trisubstituted pyridine residue to which three of thiazole units are attached, a 2-amino-2-butanoic acid unit with an aminoacetone residue, a (*Z*)-2-amino-2-butenoic acid unit attached to a threonine residue, and a 3-hydroxyvaline moiety. There was a close similarity in structures of YM-266183 and YM-266184 except for the presence of a methoxy group (C55) in YM-266184 instead of the hydroxy group of YM-266183. These new antibacterial substances were found to exhibit activity against drug-resistant bacteria [113].

Kocurin is a new anti-methicillin-resistant *Staphylococcus aureus* (MRSA) bioactive compound, belonging to the thiazolyl peptide family of antibiotics, obtained from sponge-derived *Kocuria* and *Micrococcus* spp. Structural analysis of this thiopeptide indicated the presence of several heteroaromatic moieties, including one thiazoline and four thiazole rings, one methyloxazole ring and a 2,3,6-trisubstituted pyridine residue to which two of thiazole units and one methyloxazole unit are attached, aromatic amino acids like phenylalanine and tyrosine, and two proline units. Kocurin was found to be closely related to two known thiazolyl peptide antibiotics with similar modes of action: GE37468A and GE2270. The antimicrobial activity profile of kocurin indicated the extreme potency against Gram-positive bacteria with minimum inhibitory concentration (MIC) values of 0.25–0.5 μg/mL against methicillin-resistant *Staphylococcus aureus* (MRSA) [114].

Baringolin is a novel thiopeptide of the d series, containing a central 2,3,6-trisubstituted pyridine, derived from fermentation of the marine-derived bacterium *Kucuria* sp. The macrocycle in baringolin contained three thiazoles—a methyloxazole and pyridine ring, a thiazoline ring with an α-chiral center, and a pyrrolidine motif derived from a proline residue—in addition to three natural amino acids viz. tyrosine, phenylalanine, and asparagine. The long peptidic tail was found to be a pentapeptide containing three methylidenes resulting from dehydration of serine that is attached to the pyridine through a fourth thiazole. This thiopeptide displayed important antibacterial activity against *Staphylococcus aureus*, *Micrococcus luteus*, *Propionibacterium acnes*, and *Bacillus subtilis* at nanomolar concentrations [115].

Ala-geninthiocin, geninthiocin, and Val-geninthiocin are new broad-spectrum thiopeptide antibiotics produced from the cultured marine *Streptomyces* sp. Structural analysis of all three thiopeptides indicated the presence of heteroaromatic moieties, including one thiazole and two oxazole rings, one methyloxazole ring, and a 2,3,6-trisubstituted pyridine residue to which two of thiazole units are attached at the 2 and 3 positions, including proteinogenic amino acid viz. l-threonine. The peptide structure of Ala-geninthiocin is largely similar to geninthiocin, the only difference being the presence of an l-Alanine residue instead of dealanine at the C-terminal amide. Further, Val-geninthiocin contained l-valine moiety instead of l-hydroxyvaline of geninthiocin. Ala-geninthiocin was found to exhibit good activity against Gram-positive bacteria including *Staphylococcus aureus*, *Bacillus subtilis*, *Mycobacterium smegmatis*, and *Micrococcus luteus* as well as cytotoxicity against A549 human lung carcinoma cells. When compared to geninthiocin, Ala-geninthiocin displayed better cytotoxicity but antibiotic activity against Gram-positive bacteria was comparatively low. Val-geninthiocin was found to possess more antifungal activity against *Mucor hiemalis* and cytotoxicity against A549 human lung carcinoma cells and L929 murine fibrosarcoma in comparison to geninthiocin. Further, Ala-geninthiocin and Val-geninthiocin displayed weak to moderate antifungal activity against *Candida albicans*, whereas geninthiocin was inactive. Ala-geninthiocin and geninthiocin displayed moderate antibiotic activity against Gram-negative bacteria *Chromobacterium violaceum*, whereas val-geninthiocin was inactive [116].

### 2.5. Structural Features of Bridged Heterocyclic Peptide Bicycles

Bicyclic peptides form one of the promising platforms for drug development owing to their biocompatibility and chemical diversity to proteins. Bioactive bicyclic peptides exist as disulfide-bridged peptide bicycles (e.g., ulithiacyclamide A, B, E, F, and G), histidino-tyrosine bridged peptide bicycles (e.g., aciculitins A–C), histidino-alanine bridged peptide bicycles (e.g., Theonellamides A, B, C, F, and G and Theogrenamide) and are derived from marine sponges/tunicates, plants, and mushrooms.

Ulithiacyclamide A is a strong cytotoxic disulfide-bridged peptide bicycle characterized by a symmetrical dimeric structure consisting of oxazoline and thiazole rings in addition to a transannular disulfide isolated from marine tunicate/ascidian *Lissoclinum patella*. The structure of ulithiacyclamide B closely resembled the structure of ulithiacyclamide with the exception of the replacement of one of the two d-leucine units with d-phenylalanine residue, resulting in an asymmetrical dimeric structure. Because the configuration of both leucine and phenylalanine was *d*, both thiazole amino acids possessed *R* configurations in ulithiacyclamide. The structures of ulithiacyclamides E, F, and G are related in structure to ulithiacyclamide B but with either both (in the case of ulithiacyclamide E) or just one of the two (in the cases of ulithiacyclamides F and G) oxazoline rings existing as their hydrolyzed l-threonine counterpart. Ulithiacyclamides F and G were found to be isomers and contained one oxazoline including one “free” threonine unit and were anhydro forms of ulithiacyclamide E. Ulithiacyclamide and ulithiacyclamide B exhibited cytotoxicity against the KB cell line with IC_50_ values of 35 and 17 ng/mL, respectively [51,53,56,57,117].

Aciculitins A–C are cytotoxic and antifungal glycopeptidolipids from the lithistid sponge *Aciculites orientalis*. They consist of a bicyclic peptide structure that contains a histidine-tyrosine bridge, with an unusual combination of tyrosine and histidine residues joined through the 3′-position of tyrosine and the 5′-position of histidine [118]. Theonegramide is a peculiar antifungal peptide that presents an intra-cycle histidine-alanine bridge in which the imidazole ring is substituted by a d-arabinose moiety. The alanine portion of histidinoalanine was found to have the (*R*)-configuration while the histidine portion with the (*S*)-configuration [119]. Theonellamides (TNMs) are members of a distinctive family of sterol-binding bioactive bicyclic dodecapeptides, with theonellamide F being a novel antifungal bicyclic dodecapeptide with an unprecedented histidinoalanine bridge composed of unusual amino acid residues like *τ*-l-histidino-d-alanine, (2*S*,4*R*)-2-amino-4-hydroxyadipic acid (Ahad), and (3*S*,4*S*,5*E*,7*E*)-3-amino-4-hydroxy-6-methyl-8- (*p*-bromophenyl)-5,7-octadienoic acid (Aboa). Theonellamide F was found to be a useful agent for investigating membrane structures in cells and inhibited growth of various pathogenic fungi including *Candida* sp., *Trichophyton* sp., and *Aspergillus* sp. [120,121].

Moroidin is a unique bicyclic peptide bearing residues like histidine, tryptophan, arginine, and β-leucine, isolated from the seeds of the Chinese herb *Celosia argentea* (Amaranthaceae), that remarkably inhibited the polymerization of tubulin [122]. Celogentins are unique cyclopolypeptides containing a bicyclic ring system; an unusual C–N bond formed by Trp and His residues; and an unusual amino acid, β-substituted Leu, isolated from the seeds of *Celosia argentea*. Celogentins A–C inhibited the polymerization of tubulin, and celogentin C was found to be 4 times more potent than moroidin in the inhibitory activity [123]. Phalloidin is a rigid bicyclic peptide containing an unusual cysteine-tryptophan linkage, isolated from the death cap mushroom *Amanita phalloides*. This cycloheptapeptide is commonly used in imaging applications to selectively label F-actin in fixed cells, permeabilized cells, and cell-free experiments [124]. α-Amanitin is a highly toxic hydrophobic bicyclic octapeptide found in a genus of mushrooms known as Amanita, including *Amanita phalloides*, *Amanita verna*, and *Amanita virosa*. The cytotoxicity found in amanitin is the result of inhibition of RNA polymerases, in particular RNA polymerase II, which precludes mRNA synthesis [124].

### 2.6. Structural Features of Other Heterocyclic Peptides from Marine Resources

Azonazine is a unique anti-inflammatory peptide with a macrocyclic heterocyclic core of the benzofuro indole ring system with diketopiperazine residue and possesses structural similarity with diazonamide A. The absolute configuration of this marine sediment-derived fungus-originated complex peptide was established as 2*R*,10*R*,11*S*,19*R*. The first total synthesis of hexacyclic dipeptide ent-(−)-azonazine was accomplished using a hypervalent iodine-mediated biomimetic oxidative cyclization to construct the highly strained core [125].

The pyridine ring (in the form of 3-hydroxypicolinic acid, 3HyPic) also forms part of cyclopeptide structures such as fijimycins and etamycin. Fijimycins A–C are cyclic depsipeptides from a marine-derived *Streptomyces* sp. which possessed in vitro antibacterial activity against three methicillin-resistant *Staphylococcus aureus* (MRSA) strains. The depsipeptide fijimycin A was found to contain eight subunits including α-phenylsarcosine (l-PhSar), *N*,β-dimethylleucine (l-DiMeLeu), sarcosine (Sar), 4-hydroxyproline (d-Hyp), and 3-hydroxypicolinic acid (3HyPic). Fijimycin A was defined as a stereoisomer of etamycin A containing d-α-phenylsarcosine. While comparing the structure of fijimycin B with fijimycin A, there was disappearance of α-phenylsarcosine (PhSar) and the existence of an *N*-methylleucine (l-*N*MeLeu) residue. Comparison of structures of fijimycins C and A suggested that the alanine (Ala) moiety in fijimycin A was replaced by a serine (Ser) unit. Etamycin A, also called virifogrisein I, was isolated from cultures of a terrestrial *Streptomyces* species which exhibited considerable activity against Gram-positive bacteria as well as *Mycobacterium tuberculosis*. 

Fijimycins A and C and etamycin A exhibited strong antibiotic activities against the three MRSA strains (ATCC33591, Sanger 252, UAMS1182). However, fijimycin B showed weak inhibition against both ATCC33591 and UAMS1182, which indicated that the α-phenylsarcosine unit might be vital for significant antibacterial activity. The similar antimicrobial activities of the stereoisomers fijimycin A and etamycin A suggested that substituting d- for l-α-phenylsarcosine had little effect on the anti-MRSA activities [126].

Jaspamide P is a sponge-derived modified jaspamide derivative possessing antimicrofilament activity and characterised by a modification of the *N*-methylabrine (*N*-methyl-2-bromotrypthophan) residue. Structural analysis of this cyclopeptide indicated the presence of a 4-methoxy-1,3-benzoxazine- 2-one heteroaromatic system. Jaspamide P was found to exhibit cytotoxic activity against HT-29 and MCF-7 tumour cell lines. Modifications of the methylabrine residue, claimed as essential for the observed biological activity, appeared to have little influence on the observed antiproliferative effect [127].

Wainunuamide is an unusual histidine containing cycloheptapeptide, containing three proline units. There were adjacent *cis* and *trans* proline residues in the structure of wainunuamide. Similar patterns were also found in cyclooligopeptide phakellistatin 8 and were found to be powerful β-turn inducers. The stereochemistry of all residues including histidine, phenylalanine, and leucine was found to be *l*. Wainunuamide exhibited weak cytotoxic activity in A2780 ovarian tumor and K562 leukemia cancer cells [128].

Ohmyungsamycins A and B are marine bacterium-derived cytotoxic and antimicrobial cyclic depsipeptides composed of 12 amino acid residues, including unusual amino acids such as *N*-methyl-4-methoxy-l-tryptophan, *β*-hydroxy-l-phenylalanine, and *N,N*-dimethylvaline. Ohmyungsamycins A and B showed significant inhibitory activities against diverse cancer cells as well as antibacterial effects against both Gram-positive and Gram-negative bacteria. Sungsanpin is a serine-rich lasso peptide containing 15 amino acid units from a deep-sea streptomycete in which eight amino acids form a cyclic peptide and the remaining seven amino acids including l-tryptophan unit form a tail that loops through the ring. It is the first example of a lasso peptide from a marine-derived microorganism and displays inhibitory activity with the human lung cancer cell line A549 in a cell invasion assay [129].

Desotamide and destolamide B are l-tryptophan containing bioactive peptides from marine microbe *Streptomyces scopuliridis* SCSIO ZJ46. These cyclohexapeptides displayed good antibacterial activities against *Streptococcus pnuemoniae*, *Staphylococcus aureus*, and *methicillin-resistant Staphylococcus epidermidis* (MRSE). In a complementary fashion, the antibacterial activities of destolamides revealed the “Tryptophan” moiety to be essential, thereby highlighting a critical structural element to this advancing antibacterial scaffold [130].

## 3. Stereochemical Aspects

Stereochemistry includes the study of the relative arrangement of atoms or groups in a molecule in three-dimensional space and its understanding is crucial for the study of complex molecules like heterocyclic peptides, which are of paramount biological significance.

*cis,cis*- and *trans,trans*-ceratospongamides (**44**,**45**) are new bioactive thiazole-containing cyclic heptapeptides from the marine red alga *Ceratodictyon spongiosum* and symbiotic sponge *Sigmadocia symbiotica*. The structures of ceratospongamides (**44**,**45**) contained two l-phenylalanine residues, one (l-isoleucine)-l-methyloxazoline residue, one l-proline residue, and one (l-proline)thiazole residue and were found to be proline amide conformers. The change in conformation of a cyclooligopeptide ceratospongamide from “*trans*” to “*cis*” resulted in complete loss of bioactivity, e.g., *trans, trans*-isomer of ceratospongamide (**45**) was found to be a potent inhibitor of the expression of a key enzyme in the inflammatory cascade, secreted phospholipase A_2_ (sPLA_2_), with an ED_50_ of 32 nM in a cell-based model for anti-inflammation, whereas *cis,cis*-isomer (**44**) was inactive [77] (Figure 15).

Ulithiacyclamides are thiazole-containing cyclopolypeptides, isolated from the ascidian *Lissoclinum patella*. Bicyclic isomeric ulithiacyclamides F and G contained one oxazoline and one “free” threonine and were found to be anhydro forms of ulithiacyclamide E. Ulithiacyclamides F and G exhibited anti-multiple drug resistant (MDR) activity against vinblastine-resistant CCRF-CEM human leukemic lymphoblasts [51].

Lissoclinamides 4, 5, 7, and 8 are all cyclic heptapeptides derived from sea squirt *Lissoclinum patella* that have the same sequence of amino acids around the ring and differ from one another only in their stereochemistry or the number of thiazole and thiazoline rings. For lissoclinamide 8, the valine residue was at position 31, the same sequence that occurs in lissoclinamide 4. Therefore, the only difference between lissoclinamides 4 and 8 resided in the stereochemistry of one or two of the amino acids. The d configuration was assigned to “Phe-Tzl” and the l-configuration was assigned to “Val-Tzn” moiety in lissoclinamide 4. However, both lissoclinamides 4 and 8 contained similar residues like l-Pro-mOzn and l-Phe. Further, there was similarity in the structural components of lissoclinamides 2 and 3; the only difference was in the stereochemistry around Ala-Tzl moiety, *d* in the case of the former and *l* in the latter [55,56].

Lyngbyabellins are thiazole-containing halogenated peptolides derived from cyanobacteria, possessing cytotoxic properties. The configurations at C-15 and C-16 in lyngbyabellin A were found to be 15*S* and 16*S*. Further, C-26 and C-3 in the peptolide has the *S* configuration. The stereochemical assignments of lyngbyabellins E and H were found to be 2*S*, 3*S*, 14*R*, 20*S*, 26*R*, and 27*S*. The stereoconfigurations assigned to lyngbyabellin N was 2*S*, 3*S*, 14*R*, and 20*S*. The absolute configuration of the *N,N*-dimethylvaline (DiMeVal) residue in lyngbyabellin N was found to be *l*, whereas the absolute configurations of the leucine statine were determined to be 3*R* and 4*S*. The absolute configurations of lyngbyabellin J were found to be 2*S*, 3*S*, 14*R*, 20*R*, 21*S*, 27*R*, and 28*S*. An overall cyclic constitution was not required for potent cytotoxic properties in lyngbyabellins as acyclic peptides like lyngbyabellins F and I also exhibited significant cytotoxic properties [27,28,29,30].

The cyclopolypeptides bistratamides M and N (**46**,**47**) were found to be isomers of each other and differed in the configuration of alanine residue attached to the thiazole ring. The configuration was *l* in bistratamide M (**46**) and was found to be *d* in bistratamide N (**47**). Bistratamide M (**46**) was found to be slightly more cytotoxic against lung, breast, and pancreatic carcinoma cells in comparison to bistratamide N (**47**). Similarly, bistratamides K and L (**50**,**51**) are isomers, differing in the configuration of alanine residue attached to the thiazole ring. The configuration was *d* in bistratamide K (**50**) and was found to be *l* in bistratamide L (**51**). Further, bistratamide G (**39**) was found to be *O*-isostere of bistratamide H (**40**) and bistratamide J was found to be *S*-isostere of bistratamide I (**41**). The compounds containing two thiazole rings were found to be more active than those containing a thiazole ring and an oxazole ring [50,61]. Moreover, the gross structure of cytotoxic cyclopeptide keramamide G (**49**) was found to be almost the same as that of keramamide F (**48**), the only change being the different stereochemistry at C-13 of the α-keto-β-amino acid (Figure 16).

Grassypeptolides D and E are diasteromeric cyclic peptides from a red sea *Leptolyngbya* cyanobacterium. These cyclodepsipeptides were found to contain two aromatic residues, phenyllactic acid (Pla), *N*-methylphenylalanine (*N*-Me-Phe); *β*-amino acid residue 2-methyl-3-aminobutyric acid (Maba); and 2-aminobutyric acid (Aba) residue. Further, structural analysis indicated the presence of a 2-methylthiazoline carboxylic acid derived from *N*-methylphenylalanine (*N*-Me-Phe-4-Me-thn-ca) and an Aba-thn-ca unit. Grassypeptolides D and E showed significant cytotoxicity to HeLa (IC_50_: 335 and 192 nM) and mouse neuro-2a blastoma cells (IC_50_: 599 and 407 nM). These depsipeptides were found to be threonine/N-methylleucine diastereomers and possesssed different configurations for both l-Thr and *N*-Me-l-Leu in grassypeptolide E (**53**) relative to grassypeptolide D (**52**). Grassypeptolide D (7*R*,11*R*; d-*allo*-Thr and *N*-Me-d-Leu) (**52**) was found to be approximately 1.5-fold less cytotoxic to HeLa cervical carcinoma and neuro-2a mouse blastoma cells than grassypeptolide E (7*S*,11*S*; l-Thr and *N*-Me-l-Leu) (**53**). Moreover, grassypeptolides A and C were found to be the *N*-methylphenylalanine epimers with stereochemistry (7*R*,11*R*,25*R*,29*R*) and (7*R*,11*R*,25*R*,29*S*), respectively. Grassypeptolide C showed 16–23-fold greater potency than grassypeptolide A against colorectal adenocarcinoma HT29 and cervical carcinoma HeLa cells [25] (Figure 17).

Nostocyclamide M (**54**) and tenuecyclamide C (**55**) were found to be diasteromers. Nostocyclamide M (**54**) has the same constitution as tenuecyclamide C (**55**) but differs in the configuration of methionine in the structure. Adjacent to one of thiazole ring, d-methionine was present in cyclic hexapeptide nostocyclamide M (**54**) wheresas there was l-methionine in cyclic hexapeptide tenuecyclamide C (**55**). Nostocyclamide M (**54**) displayed allelopathic activity like nostocyclamide but was inactive against grazers unlike the latter [36] (Figure 18).

Ulongamides (**1**–**3**) are thiazole-containing cytotoxic cyclic depsipeptides with a novel β-amino acid, 3-amino-2-methylhexanoic acid (Amha), stereochemistry which differentiates ulongamides A–C from ulongamides D–F. The former has the Amha residue in 2*R*,3*R* configuration, while the latter contains an Amha unit in 2*S*,3*R* configuration. The 2-hydroxy-3-methylpentanoic acid (Hmpa) residue was found to be part of ulongamide E and F (**3**) structures, and the configuration of the residue was 2*S*,3*S*. Furthermore, stereochemistry of the 2-hdroxyisovaleric acid (Hiva) unit present in ulongamide d (**2**) was found to be *S* [13].

Calyxamides A and B (**56**,**57**) are cyclic peptides containing 5-hydroxytryptophan (Htrp), isolated from the marine sponge *Discodermia calyx*. These peptides contained residues like 2,3-diaminopropionic acid (Dpr) in addition to (*O*-methylseryl)thiazole moiety. Calyxamides A and B (**56**,**57**) possessed the same planar structure but are isomeric at the 3-position of the 3-amino-2-keto-4-methylhexanoic acid (AKMH) residue like keramamides F and G (13*S* and 13*R*). Structures of calyxamides differ in stereochemistry on isoleucine moiety adjacent to (*O*-methylseryl)thiazole moiety. Calyxamide B (**57**) was found to be the diastereomer of calyxamide A (**56**) and displayed more cytotoxicity against P388 murine leukemia cells, with an IC_50_ value of 0.9 μM, in comparison to calyxamide A (IC_50_: 3.9 μM) (**56**) [110] (Figure 19).

Aciculitamides A and B are bicyclic *E* and *Z* isomeric peptides obtained from the lithistid sponge *Aciculites orientalis* and result from oxidation of the imidazole ring of aciculitins A–C, bicycles containing an unusual histidino-tyrosine bridge. Aciculitamide A did not show any cytotoxicity against HCT-116 and/or antifungal activity [118].

Sclerotides A and B are cyclopolypeptides from marine-derived fungus, *Aspergillus sclerotiorum* PT06-1. These cyclic hexapeptides contained amino acid residues like l-threonine, l-alanine, phenylalanine, serine, anthranilic acid (AA), and dehydrotryptophan (∆-Trp). Sclerotides A and B were found to be *Z* and *E* isomers and differed in stereochemistry of dehydrotryptophan. Sclerotide B showed more antifungal activity against *Candida albicans* with MIC values of 3.5 µM in comparison to sclerotide A (MIC: 7 µM). In addition, sclerotide B exhibited weak cytotoxic activity against the HL-60 cell line (IC_50_: 56.1 µM) and selective antibacterial activity against *Pseudomonas aeruginosa* (MIC: 35.3 µM) [131].

## 4. Synthesis of Heterocyclic Peptides

Despite of lot of challenges associated with synthesizing complex peptide molecules [132,133,134,135], syntheses of diverse aromatic/heteroaromatic peptides were accomplished by several research groups employing diverse techniques of peptide synthesis including solid-phase peptide synthesis (SPPS), liquid-phase peptide synthesis (LPPS), and a mixed solid-phase/solution synthesis strategy, irrespective of whether these congeners belong to linear analogues [136,137,138,139,140,141,142,143,144,145,146,147,148,149,150,151] or are cyclic in nature [152,153,154,155,156,157,158,159,160,161,162,163,164,165,166,167,168,169]. Literature is enriched with reports involving synthesis of various heterocyclic cyclopolypeptides bearing thiazole/thiazoline/tryptophan/histidine moieties viz. cyclodidemnamide B [42], dolastatin 3 [90], aeruginazole A [170], didmolamide B (**29**) [171], dolastatin 10 (**20**) [172], scleritodermin A (**10**) [173], obyanamide (**8**) [174,175], marthiapeptide A [176], diandrine C [177], diandrine A [178], sarcodactylamide [179], segetalin C [180], segetalin E [181], annomuricatin B [182], and gypsin D [183].

The first total synthesis of thiazole and methyloxazoline-containing cyclohexapeptides didmolamides A and B was accomplished by the solid phase assembly of thiazole-containing amino acids and Fmoc-protected α-amino acids. The synthesis of thiazole-containing didmolamide B (**29**) was also achieved using solution phase peptide synthesis. The crucial thiazole amino acid was synthesized by MnO_2_ oxidation of a thiazoline prepared from an Ala-Cys dipeptide using bis(triphenyl)oxodiphosphonium trifluoromethanesulfonate. The final macrolactamization was accomplished efficiently by benzotriazole-1-yl-oxy-tris-pyrrolidino-phosphonium hexafluoro- phosphate (PyBOP) and 4-dimethylaminopyridine (DMAP) [171].

A practical approach to asymmetric synthesis of dolastatin 10 (**20**) was found to involve SmI2-induced cross-coupling and asymmetric addition of chiral N-sulfinyl imine [172].

The synthesis of the C1–N15 fragment of the marine natural product scleritodermin A (**10**) was accomplished through a short and stereocontrolled sequence. The highlights of this route included synthesis of a novel conjugated thiazole moiety 2-(1-amino-2-*p*-hydroxyphenylethane)-4- (4-carboxy-2,4-dimethyl-2Z,4E-propadiene)-thiazole (ACT) fragment and the formation of the α-keto amide linkage by the use of a highly activated α,β-ketonitrile [173]. The total synthesis of a cytotoxic *N*-methylated thiazole-containing cyclic depsipeptide obyanamide (**8**) was accomplished that included the preparation of two protected fragments before macrocyclization, starting from material (S)-2-aminobutyric acid. The synthesis has led to a reassignment of the C-3 configuration in β-amino acid residue. As a result, the configuration at C-3 position has been amended as *R* [174,175]. 

The cytotoxic polythiazole-containing cyclopeptide marthiapeptide A having a linked trithiazole−thiazoline system was synthesized via two routes. The initial strategy involved a macrocyclization of the linear precursor via a peptide-coupling reaction between the amine on the alanine residue and the carboxylic acid end of isoleucine. However, the cyclization was not successful, which was attributed to the closing point being too close to the rigid heterocyclic thiazole moiety. The second strategy involved closing between the thiazoline and peptide in which successful cyclization can be attributed to the flexibility of the thiazoline, which allows a connection between the molecule’s head and tail [176].

## 5. Structural Activity Relationships

Structural activity relationships (SAR) are prime keys to diverse aspects of drug discovery, ranging from primary screening to extensive lead optimization. SAR can be used to predict bioactivity from the molecular structure. This powerful technology is used in drug discovery to guide the acquisition or synthesis of desirable new compounds as well as to further characterize existing molecules. The principle of structure–activity relationship indicated that there is a relationship between molecular structures and their biological activity and solely depends on the recognition of which structural characteristics correlate with chemical and biological reactivity.

The lissoclinamides, heterocyclic peptides isolated from sea squirt Lissoclinum patella, are derived from a cyclic heptapeptide in which a threonine has been cyclised to an oxazoline and two cysteines have been cyclised to give a thiazole or thiazoline. While comparing natural and synthetic lissoclinamides, it was found that the replacement of thiazoline rings with oxazolines decreased activity to a greater extent than replacement of oxazoline rings with thiazolines [184]. This study further showed that it was not the individual components of the macrocycle that conferred high activity, but rather, the overall conformation of this molecule was responsible for the bioactivity. While comparing structures of lissoclinamides 4 and 5, it was observed that these compounds differ only in the oxidation state of a single thiazole unit but that this difference makes lissoclinamide 5 two orders of magnitude less cytotoxic than lissoclinamide 4 against bladder carcinoma (T24) cells [55].

In raocyclamides (**42**,**43**), the presence of oxazoline moiety was found to be essential for cytotoxicity against sea urchin embryos. The cyanobacterium-derived cyclopolypeptides raocyclamide A and B (**42**,**43**) possessed thiazole and oxazoline rings in their composition, but raocyclamide A (**42**) contained an additional oxazoline moiety in its structure. This structural change results in a lot of variation in the biological response. While comparing the bioeffects of these cyclopolypeptides, it was found that raocyclamide A (**42**) inhibited the division of embryos of *Paracentrotus lividus* with an effective dose for 100% inhibition (ED_100_) of 30 µg/mL, whereas raocyclamide B (**43**) was inactive even at the concentrations of 250 µg/mL [32].

Replacement of d-valine moiety with d-methionine adjacent to one of the thiazole rings in the structure of macrocyclic thiazole and methyloxazole-containing allelochemical nostocyclamide resulted in cyanobacterial cyclopeptide nostocyclamide M (**54**) with inactivity toward grazers, but this structural modification does not affect the allelopathic activity against anabaena 7120 [36].

The reduction of isoleucylthiazole (Ile-Tzl) residue of a thiazole- and methyloxazoline-containing cyclooligopeptide of cyanobacterial origin, aerucyclamide B, to an isoleucylthiazoline (Ile-Tzn) residue resulted in a close analogue aerucyclamide A. From this one structural modification, the antiplasmodial activity was found to decrease by 1 order of magnitude. Further, the cyclohexapeptide aerucyclamide C underwent hydrolysis reaction using trifluoroacetic acid to form ring-opened products microcyclamide 7806A and microcyclamide 7806B. This change in structure from rigid, disk-like cyclamides to methyloxazoline (mOzn) ring-opened hydrolysis products resulted in loss of antimicrobial and cytotoxic activities [38]. In comparison, aerucyclamide B was the most active antiplasmodial compound among aerucyclamides against chloroquine-resistant strain K1 of *P. falciparum*, with selectivity against a rat myoblast cell line, whereas against parasite T. brucei rhodesiense, the most active compound was aerucyclamide C.

The cyclic structure of oxazole-rich, thiazole-containing polypeptide mechercharmycin A was found to be essential for its strong antitumor activity against human lung cancer and leukemia cells. The cyclic ring opening of mechercharmycin A resulted in linear peptide mechercharmycin B which did not displayed any inhibitory activity toward any of the cell lines [79].

The ascidian-derived cytotoxic cyclic hexapeptides, bistratamides A and B, differed from each other only by the presence or absence of one double bond. The conversion of one thiazoline in bistratamide A to a thiazole in bistratamide B, i.e., oxidation of thiazoline to thiazole, resulted in a less toxic compound. For example, comparing bioactivities of bistratamides A and B, the former has an IC_50_ value of about 50 µg/mL and latter has an IC_50_ value greater than 100 µg/mL against human cell lines including fibroblasts and bladder carcinoma cells [60].

Replacement of the alanine unit adjacent to the thiazole ring by a threonine unit in cyanobacterium-derived modified cyclohexapeptide venturamide A (**31**) resulted in a related cyclic hexapeptide venturamide B (**32**). This structural change reflected an increase in antimalarial activity against *Plasmodium falciparum* and cytotoxic activity toward mammalian Vero cells. However, with this modification, a decrease in bioactivity against *Trypanasoma cruzi* and MCF-7 cancer cells was observed [34].

The lyngbyabellin family of thiazole-containing peptolides are known to exhibit moderate to potent cytotoxicity against a number of different cancer cell types through the promotion of actin polymerization. In the HCT116 colon cancer cell line assay, reproducible IC_50_ values (40.9 ± 3.3 nM) were obtained for lyngbyabellin N, confirming the potent cytotoxic effect of this new member of the lyngbyabellin class and suggesting that the side chain of lyngbyabellin N was an essential structural feature for this potent activity. However, this trend was not entirely consistent within this structure class as other lyngbyabellin analogs lacking the side chain were found to exhibit bioactivity against HT29 and HeLa cells [29]. When compared to lyngbyabellin A, lyngbyabellin J displayed slightly less bioactivity against HT29 colorectal adenocarcinoma and HeLa cervical carcinoma cells. The cytoskeletal actin-disrupting lyngbyabellin 27-deoxylyngbyabellin A was found to be more potent than lyngbyabellin A against HT29 and HeLa carcinoma cell lines (IC_50_ values: 27-deoxylyngbyabellin A, 0.012 and 0.0073 µM; lyngbyabellin A, 0.047 and 0.022 µM), indicating the importance of hydroxylation at the C-27 position. However, lyngbyabellin A, its 27-deoxy analog, and lyngbyabellin J exhibited more cytotoxic activity against the two cell lines when compared to peptolide lyngbyabellin B (IC_50_ values: 1.1 and 0.71 µM). The configuration of the hydroxy acid-derived unit esterified to the 7,7-dichloro-3-acyloxy-2-methyloctanoic acid residue (here, Dhmpa) was not found to have a profound effect on the activity. Furthermore, close analysis of bioactivity data indicated that the cytotoxicity of cyclic and acyclic lyngbyabellins appeared to be similar [30].

The antithrombin cyclopolypeptides and cyclotheonellazoles had structural features similar to another *Theonella* sponge-derived peptide oriamide (**9**) in having nonproteinogenic amino acids like 4-propenoyl-2-tyrosylthiazole and 3-amino-4-methyl-2-oxohexanoic acid and showed potent inhibitory activity against the serine protease enzymes chymotrypsin and elastase. Cyclotheonellazole complexes with elastase/chymotrypsin exhibit a tetrahedral transition state involving the keto group of Amoha and Ser195 of elastase, while the side chain of Amoha fits in the enzyme S1 pocket. Cyclotheonellazole A, which contains a 2-aminopentanoic acid residue, was found to be the most potent inhibitor. This was probably due to a better compatibility with the enzyme S2 subsite. Cyclotheonellazoles B and C contained the amino acids leucine and homoalanine, and it appeared that the length and the branching of the aliphatic chain influenced the bioactivity. Further, these cyclopeptides were inactive against the malaria parasite plasmodium falciparum at IC_50_ values of greater than 20 μg/mL [68].

Ulongamides (**1**–**3**) are cyanobacterium-derived β-amino acid- and thiazole-containing cyclic peptides with weak cytotoxic properties. In cyclodepsipeptide ulongamide F (**3**), the lack of an aromatic amino acid or the N-methyl group adjacent to the hydroxyl acid (N-methylphenylalanine/N-methyl tyrosine in ulongapeptides A–E and l-valine in ulongapeptide F) was found to be detrimental to bioactivity. This was evident from the observation that ulongamide F (**3**) was inactive at <10 µM against KB and LoVo cells in comparison to ulongapeptides A (**1**) and D (**2**), which displayed cytotoxicity against both cell lines [13].

## 6. Biological Activity

Although thiazole-containing cyclopolypeptides of marine origin are associated with a number of bioactivities including antitubercular, antibacterial, antifungal, and inhibitory activity against serine protease enzymes chymotrypsin and elastase; anti-HIV activity; antiproliferative activity; antimalarial activity; and inhibitory activity against the transcription factor activator protein-1, the majority of them were found to exhibit anticancer activity. Various pharmacological activity-associated marine-derived Tzl-containing cyclopolypeptides along with susceptible cell line/organism with minimum inhibitory concentration are tabulated in Table 2.

## 7. Mechanism of Action

Heterocyclic thiazole-based peptides act by a variety of mechanisms including inhibiting microtubule assembly/mitosis, arresting nuclear division, inducing tumor cell apoptosis, causing microtubule depolymerization, inhibiting the protein secretory pathway through preventing cotranslational translocation, inducing G1 cell cycle arrest and an apoptotic cascade, inhibiting the phosphorylation of ERK and Akt, disrupting the cellular actin microfilament network, overproducing 1,3-β-D-glucan, activating the caspase-3 protein expression and decrease in B-cell lymphoma 2 (Bcl-2) levels, inhibiting nuclear factor kappa-light-chain-enhancer of activated B cells (NF-κB) luciferase and nitrite production, etc.

Dolastatin 10 (**20**) is a pentapeptide with potential antineoplastic activity, derived from marine mollusk Dolabella auricularia. Its mechanism of action involves the inhibition of tubulin polymerization, tubulin-dependent guanosine triphosphate hydrolysis, and nucleotide exchange, and it is a potent noncompetitive inhibitor of vincristine binding to tubulin. Binding to tubulin, dolastatin 10 (**20**) inhibits microtubule assembly, resulting in the formation of tubulin aggregates and inhibition of mitosis. This thiazole-containing linear peptide also induces tumor cell apoptosis through a mechanism involving bcl-2, an oncoprotein that is overexpressed in some cancers. Microtubule inhibitors from several chemical classes can block the growth and development of malarial parasites, reflecting the importance of microtubules in various essential parasite functions. Dolastatin 10 (**20**) was a more potent inhibitor of *P. falciparum* than any other microtubule inhibitor like dolastatin 15. Dolastatin 10 (**20**) caused arrested nuclear division and apparent disassembly of mitotic microtubular structures in the parasite, indicating that compounds binding in the “Vinca domain” of tubulin can be highly potent antimalarial agents [185].

Symplostatin 1 (**21**), an analog of dolastatin 10 (**20**), is a potent antimitotic with antiproliferative effects that act by causing microtubule depolymerization, formation of abnormal mitotic spindles that lead to mitotic arrest, and initiation of apoptosis involving the phosphorylation of the anti-apoptotic protein Bcl-2. Symplostatin 1 (**21**) inhibited the polymerization of tubulin in vitro, consistent with its mechanism of action in cells and suggesting that tubulin may be its intracellular target. Additionally, symplostatin 1 (**21**) was found to inhibit the proliferation and migration of endothelial cells, suggesting that it may have antiangiogenic activity [186].

Largazole is a cyclic peptide with thiazole/thiazoline residues, including a number of unusual structural features, including a 3-hydroxy-7-mercaptohept-4-enoic acid unit and a 16-membered macrocyclic cyclodepsipeptide skeleton. Largazole showed potent and highly selective inhibitory activities against class I HDACs (histone deacetylases) and displayed superior anticancer properties. Largazole was found to strongly stimulate histone hyperacetylation in the tumor, showed efficacy in inhibiting tumor growth and induced apoptosis in the tumor. This effect is likely mediated by modulation of levels of cell cycle regulators, by antagonism of the AKT pathway through IRS-1 downregulation, and by reduction of epidermal growth factor receptor levels [187].

Lyngbyabellins are hectochlorin-related peptides with thiazole moieties that are associated with actin polymerization activity. These lipopeptides were found to induce perceptible thickening of the cytoskeletal elements with a relatable increase in binucleated cells. Lyngbyabellin A was found to disrupt the cellular actin microfilament network in A10 and, accordingly, disrupted cytokinesis in colon carcinoma cells, causing the formation of apoptotic bodies. Lyngbyabellin E exhibited actin polymerization ability and was found to completely block the cellular microfilaments, forming binucleated cells [188].

Scleritodermin A (**10**) is a cytotoxic cyclic peptide with an unusual N-sulfated side chain and a novel conjugated thiazole moiety as well as an α-ketoamide group. Scleritodermin A (**10**) has significant in vitro cytotoxicity against a panel of human tumor cells lines, and this depsipeptide acts through inhibition of tubulin polymerization and the resulting disruption of microtubules, which is the target of a number of clinically useful natural product anticancer drugs [64]. 

Theonellamides are sponge-derived antifungal and cytotoxic bicyclic dodecapeptides with a histidine-alanine bridge. Specific binding of these peptides to 3β-hydroxysterols resulted in overproduction of 1,3-β-D-glucan and membrane damage in yeasts. The inclusion of cholesterol or ergosterol in phosphatidylcholine membranes significantly enhanced the membrane affinity of theonellamide A because of its direct interaction with 3β-hydroxyl groups of sterols. Membrane action of theonellamide A proceeds via binding to the membrane surface through direct interaction with sterols and modification of the local membrane curvature in a concentration-dependent manner, resulting in dramatic membrane morphological changes and membrane disruption. Theonellamides represents a new class of sterol-binding molecules that induce membrane damage and activate Rho1-mediated 1,3-beta-D-glucan synthesis [189].

Phalloidin is a tryptophan containing bicyclic phallotoxin, which functions by binding and stabilizing filamentous actin (F-actin) and effectively prevents the depolymerization of actin fibers. Due to its tight and selective binding to F-actin, derivatives of phalloidin-containing fluorescent tags are used widely in microscopy to visualize F-actin in biomedical research. Though phallotoxins are highly toxic to liver cells, they add little to the toxicity of ingested death cap, as they are not absorbed through the gut [190].

Jaspamide (Jasplakinolide) is a cytotoxic cyclodepsipeptide with bromotryptophan moiety that induces apoptosis in human leukemia cell lines and brain tumor Jurkat T cels by activation of caspase-3 protein expression and decrease in Bcl-2 levels. Apoptosis induced by Jaspamide was associated with caspase-3 activation, decreased Bcl-2 protein expression, and increased Bax levels, suggesting that jaspamide induced a caspase-independent cell death pathway for cytosolic and membrane changes in apoptosis cells and a caspase-dependent cell death pathway for poly (ADP-ribose) polymerase (PARP) protein degradation [191].

Azonazine is a unique peptide with a macrocyclic heterocyclic core of the benzofuro indole ring system with diketopiperazine residue. This hexacyclic dipeptide displayed anti-inflammatory activity and was found to act by inhibiting NF-κB luciferase and nitrite production [192].

## 8. Issues Associated with Marine Peptides in Drug Development

Marine peptides are fascinating therapeutic candidates due to their diverse bioactivities. They demonstrate significant chemical and biological diversity for drug development including minimized drug–drug interaction, less tissue accumulation, and low toxicity. Approximately 40% of existing small molecules and 70% of new candidates under development pipelines suffer from the low solubility problem, which is a major reason for their suboptimal drug delivery as well as failures in their development process. Approaches such as cyclodextrin complexation and solid dispersions have been employed to address this challenge and recommend the better formulation over their existing dosage forms [193,194,195,196,197,198]. Likewise, peptides, being biomacromolecules, also exhibit various challenges such as limited water solubility, stability aspects, as well as structural and synthesis complexities, limiting their full exploitation in drug development [199,200]. Table 3 portrays various issues associated with peptide drug development. Amidst the major challenges, difficulty in optimization of the required peptide length to achieve pharmacologically useful levels for receptor activation accounts for the hindered drug development of marine-based peptides. The optimization depends on variables including the size, accessibility, and fit of ligand-binding surfaces, ligand stability, and receptor residency time. Further, the high proteolytic instability of peptide-based therapeutics can be conquered by alteration of the side chains and amide bonds, which in turn makes the peptide resistant to proteolytic degradation [201]. The challenges of low bioavailability and short half-life can be overpowered by three approaches: (i) modification of the peptide backbone through the introduction of D-amino acids or unnatural amino acids, (ii) alteration of the peptide bonds with reduced amide bonds or β-amino acids, and (iii) attachment of a fatty acid. Approaches (i) and (ii) drive the peptide backbone through introducing cyclization, reduced flexibility, and enzyme digestion. Approach (iii) could lead to more specific binding to the target leading to enhanced half-life and bioavailability with fewer side effects [202]. Intracellular delivery of peptides has been a subject of interest due to their membrane-binding ability to exert action on the cell surface. Also, involving the protein transduction domain allows intracellular peptide delivery. Although the liposomal and nanoparticle drug delivery takes advantage of fusing the peptides for intracellular drug delivery, they also face the problem of low encapsulation efficiency [202]. During the process development of peptide synthesis, it is difficult to identify the critical process parameters to achieve expected purity and yield. In addition, the peptide synthesis process also depends on the specifications or requirements and targeted volumes. However, the establishment of acceptable standards and proven ranges may be lacking, which in turn accelerates their manufacturing costs during drug development.

## 9. Peptide Market and Clinical Trials

As a class of drugs, peptides are increasingly important in medicine. The Food and Drug Administration (FDA) has seen a rapid increase in the number of new drug applications submitted for peptide drug products. The availability of generic versions of these products will be critical to increasing public access to these important medications. However, ensuring the quality and equivalence between generic and brand-name peptide drug products raises a number of challenges, and those challenges differ according to the type of peptide drug. For peptide drug products with a specifically defined sequence of amino acids, the challenge has been with impurities that may be inadvertently introduced during the production process that may affect a proposed generic drug’s safety profile. Peptide-related impurities can be especially difficult to detect, analyze, and control because they usually have similar sequences to the drug itself. As per the current calculations, the market for peptide and protein drugs is estimated around 10% of the entire pharmaceutical market and will make up an even larger proportion of the market in the future. Since the early 1980s, more than 200 therapeutic proteins and peptides are approved for clinical use by the US-FDA [203].

Promising preclinical data led to clinical evaluation of a thiazole-containing linear pentapeptide, dolastatin 10 (**20**), isolated from sea hare as well as cyanobacterium. The potent antimitotic compound, dolastatin 10 (**20**), was evaluated in many phase I and phase II clinical trials for solid tumor, including a multi-institutional phase II clinical trial for soft tissue sarcoma treatment [204]. Dolastatin 10 (**20**) was withdrawn from clinical trials due to adverse effects such as peripheral neurophathy in cancer patients. Dolastatin 10 (**20**) was not found to be successful in human clinical trials, but it acted as a valuable source for a number of related compounds with clinical significance like ILX651, LU103793, and soblidotin [205,206]. Chemical modification efforts to reduce toxicity resulted in the synthesis of TZT-1027 (soblidotin or auristatin PE), a microtubule-disrupting compound, which entered a phase II clinical trial in patients with advanced or metastatic soft tissue sarcomas and lung cancer. Soblidotin has not progressed further beyond phase II clinical trials due to the associated hematological toxicities [207].

Although due to poor water solubility dolastatin 15 could not enter clinical trials, the investigations on this linear depsipeptide encouraged the development of its synthetic analogs like synthadotin and cematodin which have entered clinical trials. Preclinical studies confirmed the antitumor potential of the orally active microtubule inhibitor synthadotin against padiatric sarcomas. This depsipeptide has completed three phase II trials for the treatment of hormone refractory prostate cancer and metastatic melanoma that indicated toward the favorable toxicity profiles of synthadotin [208]. Another synthetic analog of dolastatin 15, cemadotin, underwent many phase I and phase II clinical trials against metastatic breast cancer and malignant melanoma. However, clinical trials were discontinued because of inconsiderable cytotoxicity caused by cemadotin in phase II trials and to acute myocardial infarction and neutropenia in phase I clinical trials [209].

Further modifications of soblidotin/auristatin E led to the development of monomethyl auristatin E (MMAE) and monomethyl auristatin F (MMAF), each of which included a secondary amine at their N-terminus. MMAE and MMAF have been used as warheads to link monoclonal antibodies and are presently in many clinical trials for the treatment of cancer, and eventually various antibody-drug conjugates received FDA approval [210,211].

## 10. Conclusions and Future Prospects

In present times, there is an increased frequency of resistance for conventional drugs. This fact necessitates the focus of drug research to be shifted toward a new era where bioactive compounds are developed with novel mechanisms of action. TBPs with unique structural features claim their candidature to overcome the existing issues. Various bioactive heteroaromatic peptides have been isolated from different organisms ranging from marine sponges, mollusks, and tunicates to terrestrial cyanobacteria and other microbes including fungi and bacteria. On this basis, various mimetics of bioactive peptides have been synthesized using solid and solution phase techniques of peptide synthesis. Despite enormous potential, utilization of these bioactive peptides is limited due to their stability and bioavailability issues. This review portrays recent updates and future perspectives of TBPs to attract the attention of researchers and scientists leading the efforts toward their clinical translation from the bench to the bedside.

## Figures and Tables

**Figure 1 marinedrugs-18-00329-f001:**
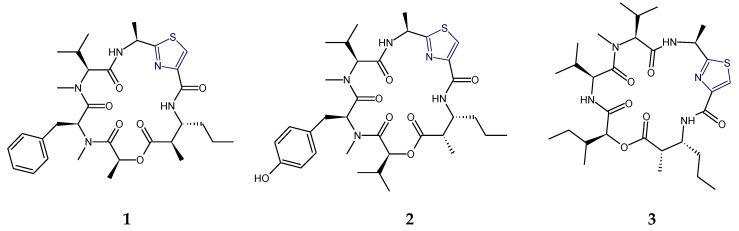
Structures of ulongamide A (**1**), ulongamide D (**2**), and ulongamide F (**3**) with alanylthiazole (Ala-Tzl) and 3-amino-2-methylhexanoic acid (Amha) moieties.

**Figure 2 marinedrugs-18-00329-f002:**
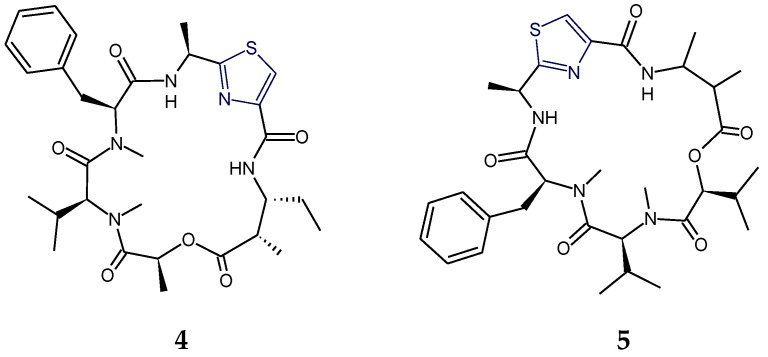
Structures of guineamide A (**4**) and guineamide B (**5**) with Ala-Tzl and l-*N*-Methylated amino acid units.

**Figure 3 marinedrugs-18-00329-f003:**
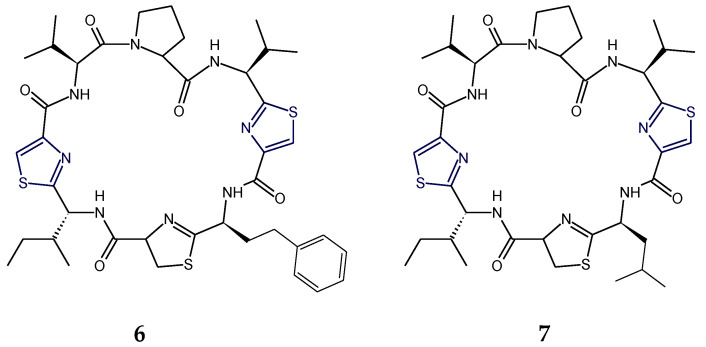
Structures of tawicyclamide A (**6**) and tawicyclamide B (**7**) with valylthiazole (Val-Tzl) and l-isoleucyl-thiazole (Ile-Tzl) moieties.

**Figure 4 marinedrugs-18-00329-f004:**
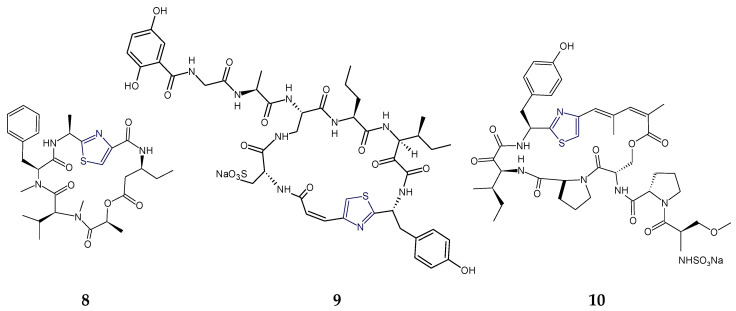
Structures of obyanamide (**8**) with Ala-Tzl moiety, oriamide (**9**) with 4-propenoyl-2-tyrosylthiazole amino acid (PTT) moiety, and scleritodermin A (**10**) with 2-(1-amino-2-*p*-hydroxyphenylethane)-4- (4-carboxy-2,4-di-methyl-2*Z*,4*E*-propadiene)-thiazole (ACT) moiety.

**Figure 5 marinedrugs-18-00329-f005:**
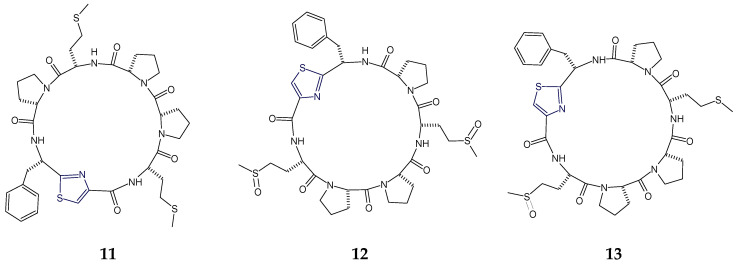
Structures of haligramide A (**11**), waiakeamide (**12**), and haligramide B (**13**) with phenylalanylthiazole (Phe-Tzl) moieties.

**Figure 6 marinedrugs-18-00329-f006:**
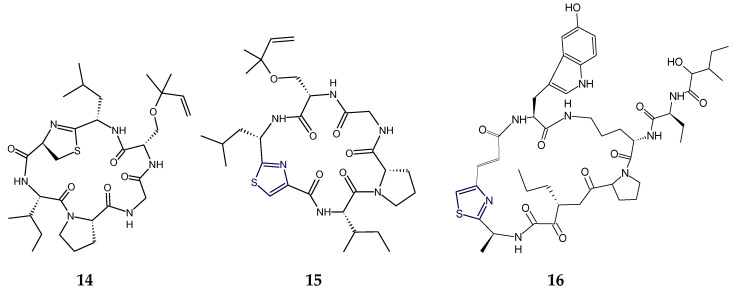
Structures of keenamide A (**14**) with leuylthiazoline (Leu-Tzn) moiety, mollamide C (**15**) with Leu-Tzl moiety, and jamaicensamide A (**16**) with Ala-Tzl and 2-hydroxy-3-methylpentanamide (Hmp) residues.

**Figure 7 marinedrugs-18-00329-f007:**
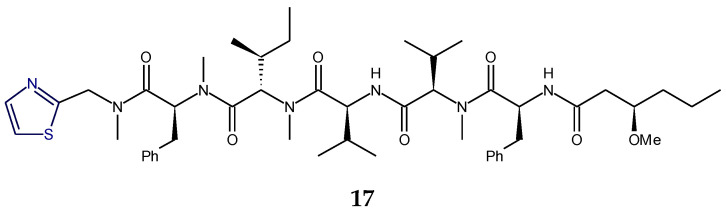
Structures of micromide (**17**), apramide A (**18**), and apramide C (**19**) with terminal *N*-Me-Gly-Tzl residues.

**Figure 8 marinedrugs-18-00329-f008:**
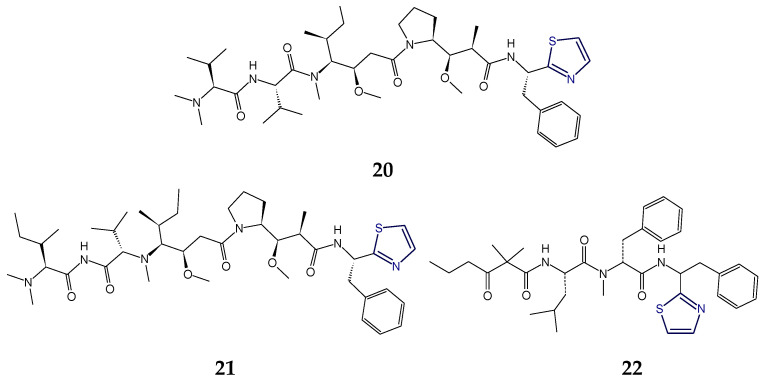
Structures of dolastatin 10 (**20**), symplostatin 1 (**21**), and dolastatin 18 (**22**) with terminal Phe-Tzl residues.

**Figure 9 marinedrugs-18-00329-f009:**
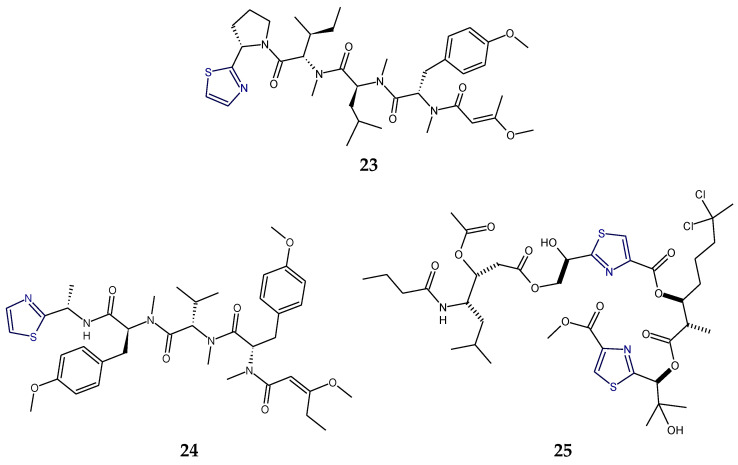
Structures of lyngbyapeptin A (**23**) with Pro-Tzl moiety, lyngbyapeptin C (**24**) withAla-Tzl moiety, lyngbyabellin F (**25**) with α,β-dihydroxyisovaleric acid (DHIV)-Tzl residue, lyngbyabellin I (**26**) with Val-Tzl moiety, and lyngbyapeptin D (**27**) with Pro-Tzl moiety.

**Figure 10 marinedrugs-18-00329-f010:**
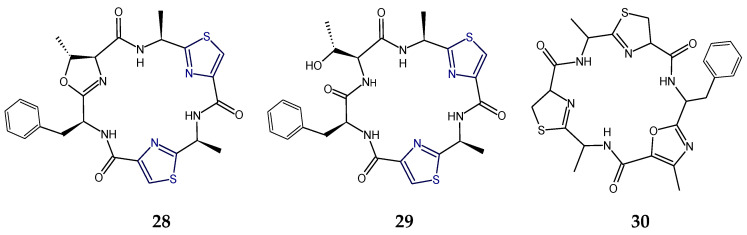
Structures of didmolamide A (**28**) with Ala-Tzl moieties, didmolamide B (**29**) with Ala-Tzl moieties, and didmolamide C (**30**) with Ala-Tzn moieties.

**Figure 11 marinedrugs-18-00329-f011:**
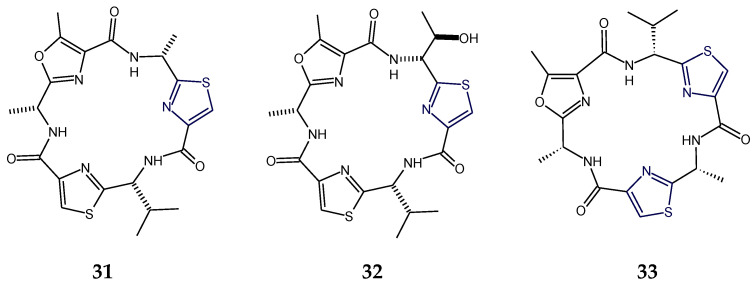
Structures of venturamide A (**31**) with Ala-Tzl and Val-Tzl residues, venturamide B (**32**) with Thr-Tzl and Val-Tzl residues, and dendroamide A (**33**) with Val-Tzl and Ala-Tzl residues.

**Figure 12 marinedrugs-18-00329-f012:**
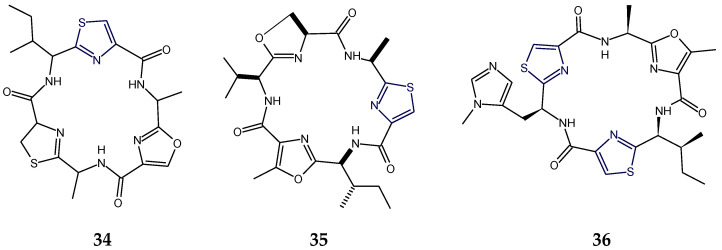
Structures of dolastatin E (**34**) with Ile-Tzl moiety, dolastatin I (**35**) with Ala-Tzl moiety, and microcyclamide (**36**) with Ile-Tzl and *N*-Me-His-Tzl residues.

**Figure 13 marinedrugs-18-00329-f013:**
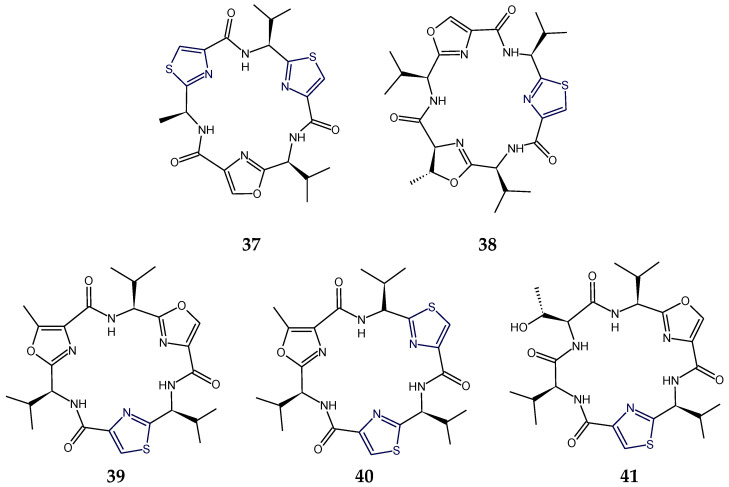
Structures of bistratamide C (**37**) with Val-Tzl and Ala-Tzl residues, bistratamide D (**38**) with Val-Tzl moiety, bistratamide G (**39**) with Val-Tzl moiety, bistratamide H (**40**) with two Val-Tzl residues, and bistratamide I (**41**) with Val-Tzl moiety.

**Figure 14 marinedrugs-18-00329-f014:**
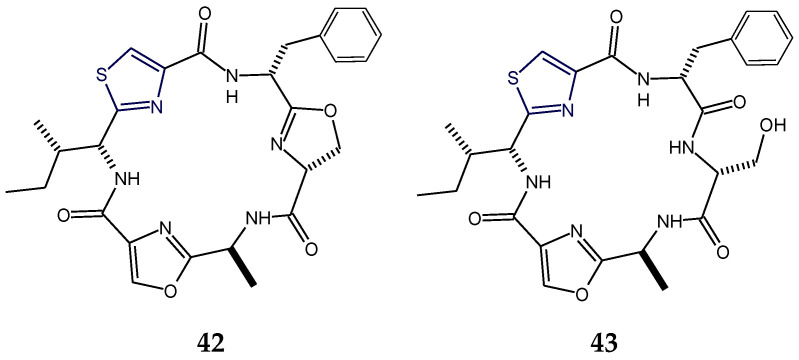
Structures of raocyclamide A (**42**) and raocyclamide B (**43**) with d-Ile-Tzl residues.

**Figure 15 marinedrugs-18-00329-f015:**
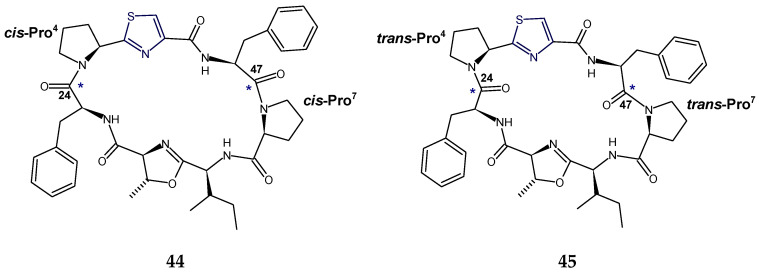
Structures of *cis,cis*-ceratospongamide (**44**) and *trans,trans*-ceratospongamide (**45**) with Pro-Tzl residues (*change in stereochemistry at C-24 and C-47 carbonyls).

**Figure 16 marinedrugs-18-00329-f016:**
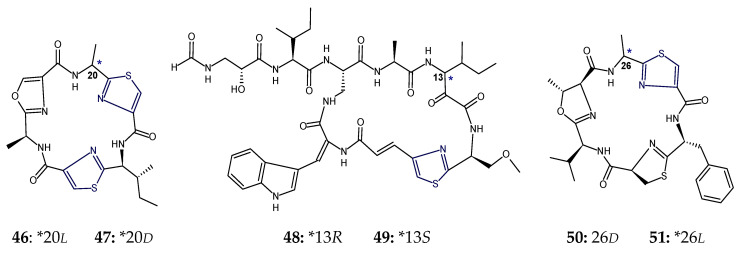
Structures of bistratamide M (**46**) with configuration *l* at C-20, bistratamide N (**47**) with configuration *d* at C-20, keramamide F (**48**) with stereochemistry *R* at C-13, keramamide G (**49**) with stereochemistry *S* at C-13, bistratamide K (**50**) with configuration *d* at C-26, and bistratamide l (**51**) with configuration *l* at C-26.

**Figure 17 marinedrugs-18-00329-f017:**
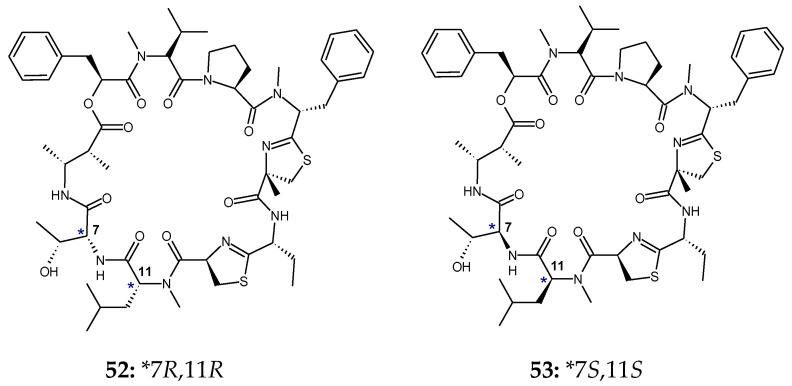
Structures of grassypeptolide D (**52**) with stereochemistry *R* at C-7 and C-11 of d-*allo*-Thr and *N*-Me-d-Leu residues and grassypeptolide E (**53**) with stereochemistry *S* at C-7 and C-11 of l-Thr and *N*-Me-l-Leu residues.

**Figure 18 marinedrugs-18-00329-f018:**
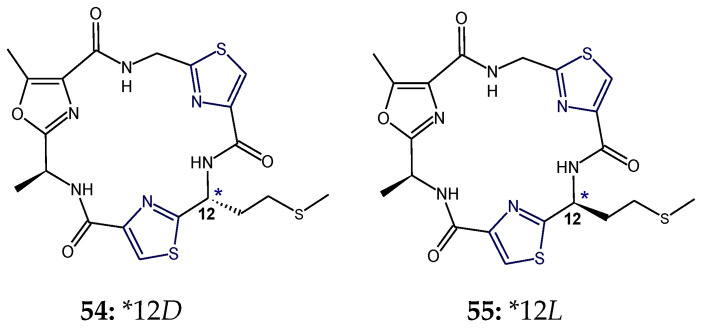
Structures of nostocyclamide M (**54**) with Gly-Tzl and Met-Tzl residues, having methionine configuration *d* at C-12, and tenuecyclamide C (**55**) with Gly-Tzl and Met-Tzl residues, having methionine configuration *l* at C-12.

**Figure 19 marinedrugs-18-00329-f019:**
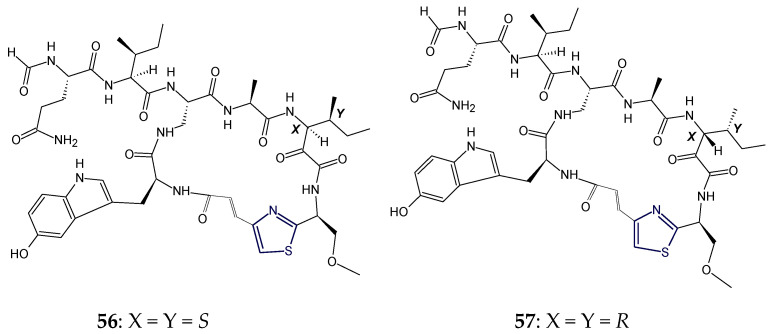
Structures of calyxamide A (**56**) with *O*-Me-Ser-Tzl moiety, having stereochemistry *S* at the 3-position of 3-amino-2-keto-4-methylhexanoic acid (AKMH) residue, and Calyxamide B (**57**) with *O*-Me-Ser-Tzl moiety, having stereochemistry *R* at the 3-position of AKMH residue.

**Table 1 marinedrugs-18-00329-t001:** Heterocyclic thiazole-based cyclopolypeptides from marine resources.

Year	Cyclic Peptide	Molecular Formula	Composition	Heterocyclic Ring (s) *
1980	Ulicyclamide [53]	C_33_H_39_N_7_O_5_S_2_	cyclooligopeptide	Tzl, mOzn
1980	Ulithiacyclamide [53]	C_32_H_42_N_8_O_6_S_4_	bicyclic peptide	Tzl, mOzn
1982	Patellamide A [39]	C_35_H_50_N_8_O_6_S_2_	cyclooctapeptide	Tzl, Ozn, mOzn
1982	Patellamide B [39]	C_38_H_48_N_8_O_6_S_2_	cyclooctapeptide	Tzl, mOzn
1982	Patellamide C [39]	C_37_H_46_N_8_O_6_S_2_	cyclooctapeptide	Tzl, mOzn
1983	Ascidiacyclamide [106]	C_36_H_52_N_8_O_6_S_2_	cyclopolypeptide	Tzl, mOzn
1989	Lissoclinamide 4 [56]	C_38_H_43_N_7_O_5_S_2_	cycloheptapeptide	Tzl, Tzn, mOzn
1989	Lissoclinamide 5 [56]	C_38_H_41_N_7_O_5_S_2_	cycloheptapeptide	Tzl, mOzn
1989	Ulithiacyclamide B [57]	C_35_H_40_N_8_O_6_S_4_	bicycle peptide	Tzl, mOzn
1989	Patellamide D [80]	C_38_H_48_N_8_O_6_S_2_	cyclooctapeptide	Tzl, mOzn
1990	Lissoclinamide 8 [55]	C_38_H_43_N_7_O_5_S_2_	cycloheptapeptide	Tzl, Tzn, mOzn
1990	Lissoclinamide 7 [55]	C_38_H_45_N_7_O_5_S_2_	cycloheptapeptide	Tzn, mOzn
1992	Tawicyclamide A [41]	C_39_H_51_N_8_O_5_S_3_	cyclooctapeptide	Tzl, Tzn
1992	Tawicyclamide B [41]	C_36_H_53_N_8_O_5_S_3_	cyclooctapeptide	Tzl, Tzn
1992	Patellamide E [58]	C_39_H_50_N_8_O_6_S_2_	cyclooctapeptide	Tzl, mOzn
1992	Bistratamide C [59]	C_22_H_26_N_6_O_4_S_2_	cyclohexapeptide	Tzl, Ozl
1992	Bistratamide D [59]	C_25_H_34_N_6_O_5_S	cyclohexapeptide	Tzl, Ozl, mOzn
1995	Keramamide J [67]	C_33_H_58_N_10_O_11_S	cyclopolypeptide	Tzl, Trp
1995	Keramamide G [67]	C_43_H_56_N_10_O_11_S	cyclopolypeptide	Tzl, Htrp
1995	Keramamide H [67]	C_43_H_57_N_10_O_12_BrS	cyclopolypeptide	Tzl, Bhtrp
1995	Cyclodidemnamide [62]	C_34_H_43_N_7_O_5_S_2_	cycloheptapeptide	Tzl, Tzn, Ozn
1995	Dolastatin E [76]	C_21_H_26_N_6_O_4_S_2_	cyclohexapeptide	Tzl, Tzn, Ozl
1995	Lissoclinamide 3 [54]	C_33_H_41_N_7_O_5_S_2_	cycloheptapeptide	Tzl, mOzn
1995	Patellamide F [54]	C_37_H_46_N_8_O_6_S_2_	cyclooctapeptide	Tzl, Ozn, mOzn
1995	Nostocyclamide [107]	C_27_H_32_N_6_O_6_S	cyclohexapeptide	Tzl, mOzl
1996	Waiakeamide [66,108]	C_37_H_49_N_7_O_8_S_3_	cyclohexapeptide	Tzl
1996	Raocyclamide B [32]	C_27_H_32_N_6_O_6_S	cyclohexapeptide	Tzl, Ozl
1996	Raocyclamide A [32]	C_27_H_30_N_6_O_5_S	cyclohexapeptide	Tzl, Ozl, Ozn
1996	Dendramide A [40]	C_21_H_24_N_6_O_4_S_2_	cyclohexapeptide	Tzl, mOzl
1996	Dendramide B [40]	C_21_H_24_N_6_O_4_S_3_	cyclohexapeptide	Tzl, mOzl
1996	Dendramide C [40]	C_21_H_24_N_6_O_5_S_3_	cyclohexapeptide	Tzl, mOzl
1997	Oriamide [65]	C_44_H_54_N_15_O_9_S_2_Na	cyclopolypeptide	Tzl
1997	Dolastatin I [75]	C_24_H_32_N_6_O_5_S	cyclohexapeptide	Tzl, mOzl, Ozn
1998	Ulithiacyclamide E [51]	C_35_H_44_N_8_O_8_S_4_	bicyclic peptide	Tzl
1998	Comoramide B [45]	C_34_H_50_N_6_O_7_S	cyclohexapeptide	Tzn
1998	Mayotamide A [45]	C_30_H_43_N_7_O_4_S_4_	cycloheptapeptide	Tzl, Tzn
1998	Mayotamide B [45]	C_29_H_41_N_7_O_4_S_4_	cycloheptapeptide	Tzl, Tzn
1998	Keramamide K [109]	C_44_H_60_N_10_O_11_S	cyclopolypeptide	Tzl, Metrp
1998	Ulithiacyclamide F [51]	C_35_H_42_N_8_O_7_S_4_	bicycle peptide	Tzl, mOzn
1998	Ulithiacyclamide G [51]	C_35_H_42_N_8_O_7_S_4_	bicycle peptide	Tzl, mOzn
1998	Comoramide A [45]	C_34_H_48_N_6_O_6_S	cyclohexapeptide	Tzn, mOzn
1998	Patellamide G [51]	C_38_H_50_N_8_O_7_S_2_	cyclooctapeptide	Tzl, mOzn
1998	Tenuecyclamide A [105]	C_19_H_20_N_6_O_4_S_2_	cyclohexapeptide	Tzl, mOzl
1998	Tenuecyclamide C [105]	C_20_H_22_N_6_O_4_S_3_	cyclohexapeptide	Tzl, mOzl
1998	Tenuecyclamide D [105]	C_20_H_22_N_6_O_5_S_3_	cyclohexapeptide	Tzl, mOzl
2000	Haligramide A [63]	C_37_H_49_N_7_O_6_S	cyclohexapeptide	Tzl
2000	Haligramide B [63]	C_37_H_49_N_7_O_7_S	cyclohexapeptide	Tzl
2000	Dolastatin 3 [9]	C_25_H_36_N_6_O_5_S_2_	cyclopentapeptide	Tzl
2000	Homodolastatin 3 [9]	C_30_H_42_N_8_O_6_S_2_	cyclopentapeptide	Tzl
2000	Lyngbyabellin A [27]	C_29_H_40_N_4_O_7_S_2_Cl_2_	cyclodepsipeptide	Tzl
2000	Lyngbyabellin B [86]	C_28_H_40_N_4_O_7_S_2_Cl_2_	cyclodepsipeptide	Tzl, Tzn
2000	Kororamide [9]	C_45_H_64_N_10_O_10_S_2_	cyclononapeptide	Tzl, Tzn
2000	Lissoclinamide 9 [52]	C_35_H_45_N_7_O_5_S_2_	cycloheptapeptide	Tzl, Tzn, mOzn
2000	Ceratospongamide [77]	C_41_H_49_N_7_O_6_S	cycloheptapeptide	Tzl, mOzn
2000	Microcyclamide [35]	C_26_H_30_N_8_O_4_S_2_	cyclohexapeptide	Tzl, mOzl, mImz
2001	Nostocyclamide M [36]	C_20_H_22_N_6_O_4_S_3_	cyclohexapeptide	Tzl, mOzl
2002	Cyclodidemnamide B [42]	C_32_H_47_N_7_O_6_S_2_	cycloheptapeptide	Tzl
2002	Obyanamide [12]	C_30_H_41_N_5_O_6_S	cyclodepsipeptide	Tzl
2002	Ulongamide A [13]	C_32_H_45_N_5_O_6_S	cyclodepsipeptide	Tzl
2002	Ulongamide D [13]	C_34_H_49_N_5_O_7_S	cyclodepsipeptide	Tzl
2002	Ulongamide E [13]	C_35_H_51_N_5_O_7_S	cyclodepsipeptide	Tzl
2002	Ulongamide B [13]	C_32_H_45_N_5_O_7_S	cyclodepsipeptide	Tzl
2002	Ulongamide C [13]	C_36_H_45_N_5_O_7_S	cyclodepsipeptide	Tzl
2002	Ulongamide F [13]	C_30_H_49_N_5_O_6_S	cyclodepsipeptide	Tzl
2002	Banyascyclamide B [11]	C_22_H_30_N_6_O_5_S_2_	cyclohexapeptide	Tzl
2002	Banyascyclamide C [11]	C_25_H_28_N_6_O_5_S_2_	cyclohexapeptide	Tzl
2002	Banyascyclamide A [11]	C_25_H_26_N_6_O_4_S_2_	cyclohexapeptide	Tzl, mOzn
2002	Leucamide A [70]	C_29_H_37_N_7_O_6_S	cycloheptapeptide	Tzl, Ozl, mOzl
2003	Guineamide A [14]	C_31_H_44_N_5_O_6_S	cyclodepsipeptide	Tzl
2003	Guineamide B [14]	C_32_H_45_N_5_O_6_S	cyclodepsipeptide	Tzl
2003	Didmolamide A [48]	C_25_H_26_N_6_O_4_S_2_	cyclohexapeptide	Tzl
2003	Didmolamide B [48]	C_25_H_28_N_6_O_5_S_2_	cyclohexapeptide	Tzl
2003	Bistratamide J [50]	C_25_H_36_N_6_O_5_S_2_	cyclohexapeptide	Tzl
2003	Bistratamide I [50]	C_25_H_36_N_6_O_5_S_2_	cyclohexapeptide	Tzl, Ozl
2003	Bistratamide H [50]	C_25_H_32_N_6_O_4_S_2_	cyclohexapeptide	Tzl, mOzl
2003	Bistratamide E [50]	C_25_H_34_N_6_O_4_S_2_	cyclohexapeptide	Tzl, mOzn
2003	Bistratamide G [50]	C_25_H_32_N_6_O_5_S	cyclohexapeptide	Tzl, Ozl, mOzl
2003	Bistratamide F [50]	C_26_H_36_N_6_O_5_S	cyclohexapeptide	Tzl, Ozn, mOzn
2003	Myriastramide C [69]	C_42_H_53_N_9_O_7_S	cyclooctapeptide	Tzl, Ozl, Trp
2003	Bistratamide B [60]	C_27_H_32_N_6_O_4_S_2_	cyclohexapeptide	Tzl, Tzn, mOzn
2004	Scleritodermin A [64]	C_42_H_54_N_7_O_10_SNa	cyclopolypeptide	Tzl
2005	Lyngbyabellin E [28]	C_37_H_51_N_3_O_12_S_2_Cl_2_	cyclodepsipeptide	Tzl
2005	Lyngbyabellin H [28]	C_37_H_51_N_3_O_11_S_2_Cl_2_	cyclodepsipeptide	Tzl
2005	Mechercharmycin A [79]	C_35_H_32_N_8_O_7_S	cyclooligopeptide	Tzl, Ozl
2006	Trichamide [17]	C_44_H_66_N_16_O_12_S_2_	cyclopolypeptide	Tzl, His
2007	Urukthapelstatin A [78]	C_34_H_30_N_8_O_6_S_2_	cyclooligopeptide	Tzl, Ozl
2007	Venturamide A [34]	C_21_H_24_N_6_O_4_S_2_	cyclohexapeptide	Tzl, mOzl
2007	Venturamide B [34]	C_22_H_26_N_6_O_5_S_2_	cyclohexapeptide	Tzl, mOzl
2008	Mollamide C [46]	C_30_H_46_N_6_O_6_S	cyclohexapeptide	Tzl
2008	Aerucyclamide B [37]	C_24_H_33_N_6_O_4_S_2_	cyclohexapeptide	Tzl, mOzn
2008	Aerucyclamide A [37]	C_24_H_34_N_6_O_4_S_2_	cyclohexapeptide	Tzl, Tzn, mOzn
2008	Aerucyclamide D [38]	C_26_H_31_N_6_O_4_S_3_	cyclohexapeptide	Tzl, Tzn, mOzn
2008	Aerucyclamide C [38]	C_24_H_32_N_6_O_5_S	cyclohexapeptide	Tzl, Ozl, mOzn
2009	Sanguinamide A [73]	C_37_H_52_N_7_O_6_S	cycloheptapeptide	Tzl
2009	Sanguinamide B [73]	C_33_H_43_N_8_O_6_S_2_	cyclooctapeptide	Tzl, Ozl
2010	Microcyclamide MZ602 [18]	C_28_H_38_N_6_O_7_S	cyclohexapeptide	Tzl
2010	Microcyclamide MZ568 [18]	C_25_H_40_N_6_O_7_S	cyclohexapeptide	Tzl
2010	Aeruginazole A [91]	C_53_H_66_N_13_O_11_S_3_	cyclododecapeptide	Tzl
2010	Lyngbyabellin J [30]	C_37_H_51_N_3_O_12_S_2_Cl_2_	cyclodepsipeptide	Tzl
2010	27-deoxylyngbyabellin A [30]	C_29_H_40_N_4_O_6_S_2_Cl_2_	cyclodepsipeptide	Tzl
2012	Aeruginazole DA1497 [8]	C_68_H_91_N_17_NaO_14_S_4_	cyclopolypeptide	Tzl
2012	Aeruginazole DA1304 [8]	C_61_H_72_N_14_NaO_13_S_3_	cyclopolypeptide	Tzl
2012	Aeruginazole DA1274 [8]	C_60_H_70_N_14_NaO_12_S_3_	cyclopolypeptide	Tzl
2012	Lyngbyabellin N [29]	C_40_H_58_N_4_O_11_S_2_Cl_2_	cyclodepsipeptide	Tzl
2012	Largazole [16]	C_29_H_38_N_4_O_5_S_3_	cyclodepsipeptide	Tzl, Tzn
2012	Marthiapeptide A [74]	C_30_H_31_N_7_O_3_S_4_	cyclooligopeptide	Tzl, Tzn
2012	Calyxamide A [110]	C_45_H_61_N_11_O_12_S	cyclooligopeptide	Tzl, Htrp
2012	Calyxamide B [110]	C_45_H_61_N_11_O_12_S	cyclooligopeptide	Tzl, Htrp
2013	Aestuaramide A [10]	C_40_H_51_N_7_O_6_S_3_	cyclopolypeptide	Tzl
2013	Aestuaramide B [10]	C_35_H_43_N_7_O_6_S_3_	cyclopolypeptide	Tzl
2013	Aestuaramide C [10]	C_40_H_51_N_7_O_6_S_3_	cyclopolypeptide	Tzl
2014	Balgacyclamide A [33]	C_25_H_37_N_6_O_5_S	cyclooligopeptide	Tzl, mOzn
2014	Balgacyclamide B [33]	C_25_H_39_N_6_O_6_S	cyclooligopeptide	Tzl, mOzn
2014	Balgacyclamide C [33]	C_28_H_37_N_6_O_6_S	cyclooligopeptide	Tzl, mOzn
2016	Jamaicensamide A [89]	C_45_H_61_N_9_O_10_S	cyclooligopeptide	Tzl, Htrp
2017	Cyclotheonellazole A [68]	C_44_H_54_N_9_O_14_S_2_Na_2_	cyclopolypeptide	Tzl
2017	Cyclotheonellazole B [68]	C_45_H_57_N_9_O_14_S_2_Na	cyclopolypeptide	Tzl
2017	Cyclotheonellazole C [68]	C_43_H_52_N_9_O_14_S_2_Na_2_	cyclopolypeptide	Tzl
2017	Bistratamide M, N [61]	C_21_H_24_N_6_O_4_S_2_	cyclohexapeptide	Tzl, Ozl

* Tzl: Thiazole, Tzn: Thiazoline, Ozl: Oxazole, Ozn: Oxazoline, mOzl: 5-methyloxazole, mOzn: 5-methyloxazoline, Htrp: 5-hydroxytryptophan, mImz: N-methylimidazole, His: histidine, Trp: tryptophan, Bhtrp: 2-bromo-5-hydroxytryptophan, Metrp: N-methyltryptophan.

**Table 2 marinedrugs-18-00329-t002:** Heterocyclic Tzl-based peptides (TBPs) with diverse pharmacological activities.

TBPs	Resource	Bioactivity	
		Susceptibilty	MIC^a^ Value
Haligramide A [63]	marine sponge *Haliclona nigra*	Cytotoxicity against A-549 (lung), HCT-15 (colon), SF-539 (CNS^b^), and SNB-19 (CNS) human tumor cell lines	5.17–15.62 μg/mL
Haligramide B [63]	marine sponge *Haliclona nigra*	Cytotoxicity against A-549 (lung), HCT-15 (colon), SF-539 (CNS), and SNB-19 (CNS) human tumor cells	3.89–8.82 μg/mL
Scleritodermin A [64]	marine sponge *Scleritoderma nodosum*	Cytotoxicity against colon HCT116, ovarian A2780, and breast SKBR3 cell lines	0.67–1.9 μM
Obyanamide [12]	marine cyanobacterium *Lyngbya confervoides*	Cytotoxicity against KB^c^ and LoVo cells	0.58 and 3.14 µg/mL
Waiakeamide [66]	marine sponge *Ircinia dendroides*	Anti-TB activity against *Mycobacterium tuberculosis*	7.8 μg/mL
Ulongamide A [13]	marine cyanobacterium *Lyngbya* sp.	Cytotoxicity against KB and LoVo cells	1 and 5 µM
Guineamide B [14]	marine cyanobacterium *Lyngbya majuscula*	Cytotoxicity against mouse neuroblastoma cell line	15 µM
Calyxamide A [110]	marine sponge *Discodermia calyx*	Cytotoxicity against P388 murine leukemia cells	3.9 and 0.9 μM
Bistratamide J [50]	marine ascidian *Lissoclinum bistratum*	Cytotoxic activity against the human colon tumor (HCT-116) cell line	1.0 µg/mL
Didmolamide A and B [48]	marine tunicate *Didemnum molle*	Cytotoxicity against several cultured tumor cell lines (A549, HT29, and MEL28)	10–20 µg/mL
Aeruginazole A [91]	freshwater cyanobacterium *Microcystis* sp.	Antibacterial activity againt *B. subtilis* and *S. albus* Cytotoxicity against MOLT-4 human leukemia cell line and peripheral blood lymphocytes	2.2 and 8.7 μM 41 and 22.5 μM
Cyclotheonellazole A, B and C [68]	marine sponge *Theonella* aff*. swinhoei*	Inhibitory activity against serine protease enzyme chymotrypsin Inhibitory activity against serine protease enzyme elastase	0.62, 2.8, and 2.3 nM 0.034, 0.10, and 0.099 nM
Microcyclamide MZ602 [18]	cyanobacterium *Microcystis* sp.	Inhibition activity of chymotrypsin	75 μM
Dolastatin 3 [9]	marine cyanobacterium *Lyngbya majuscula*	Inhibition of HIV-1 integrase (for the terminal-cleavage and strand- transfer reactions)	5 mM and 4.1 mM
Lyngbyabellin A [27]	marine cyanobacterium *Lyngbya majuscula*	Cytotoxicity against KB cells (human nasopharyngeal carcinoma cell line) and LoVo cells (human colon adenocarcinoma cell line) Cytotoxicity against HT29 colorectal adenocarcinoma and HeLa cervical carcinoma cells Cytoskeletal-disrupting effects in A-10 cells	0.03 and 0.50 μg/mL 1.1 and 0.71 μM 0.01–5.0 μg/mL
Lyngbyabellin B [86]	marine cyanobacterium *Lyngbya majuscula*	Toxicity to brine shrimp (*Artemia salina*) Antifungal activity against *Candida albicans* (ATCC 14053) in a disk diffusion assay Cytotoxicity against HT29 colorectal adenocarcinoma and HeLa cervical carcinoma cells	3.0 ppm 100 μg/disk 1.1 and 0.71 μM
Lyngbyabellin E [28]	marine cyanobacterium *Lyngbya majuscula*	Cytotoxicity against NCI-H460 human lung tumor and neuro-2a mouse neuroblastoma cells Cytoskeletal-disrupting effects in A-10 cells	0.4 and 1.2 μM 0.01–6.0 μM
Lyngbyabellin H [28]	marine cyanobacterium *Lyngbya majuscula*	Cytotoxicity against NCI-H460 human lung tumor and neuro-2a mouse neuroblastoma cells	0.2 and 1.4 μM
Lyngbyabellin N [29]	marine cyanobacterium *Moorea bouilloni*	Cytotoxic activity against HCT116 colon cancer cell line	40.9 nM
27-Deoxy- lyngbyabellin A [30]	marine cyanobacterium *Lyngbya bouillonii*	Cytotoxicity against HT29 colorectal adenocarcinoma and HeLa cervical carcinoma cells	0.012 and 0.0073 μM
Lyngbyabellin J [30]	marine cyanobacterium *Lyngbya bouillonii*	Cytotoxicity against HT29 colorectal adenocarcinoma and HeLa cervical carcinoma cells	0.054 and 0.041 μM
Raocyclamide A [32]	filamentous cyanobacterium *Oscillatoria raoi*	Cytotoxicity against embryos of sea urchin *Paracentrotus lividus*	30 μg/mL (ED_100_)^d^
Tenuecyclamide A, C and D [105]	cultured cyanobacterium *Nostoc spongiaeforme* var. *tenue*	Cytotoxicity against embryos of sea urchin *Paracentrotus lividus*	10.8, 9.0, and 19.1 μM (ED_100_)
Dolastatin I [75]	sea hare *Dolabella auricularia*	Cytotoxicity against HeLa S_3_ cells	12 μg/mL
Marthiapeptide A [74]	marine actinomycete *Marinactinospora thermotolerans* SCSIO 00652	Antibacterial activities against *Micrococcus luteus*, *Staphylococcus aureus*, *Bacillus subtilis*, and *Bacillus thuringiensis* Cytotoxicity against SF-268 (human glioblastoma) cell line, MCF-7 (human breast adenocarcinoma) cell line, NCI-H460 (human lung carcinoma) cell line, and HepG2 (human hepatocarcinoma) cancer cell line	2.0, 8.0, 4.0, and 2.0 μg/mL 0.38, 0.43, 0.47, and 0.52 μM
Keramamide G, H and J [67]	marine sponge *Theonella* sp.	Cytotoxicity against L1210 murine leukemia cells and KB human epidermoid carcinoma cells	10 µg/mL
Keramamide K [109]	marine sponge *Theonella* sp.	Cytotoxicity against L1210 murine leukemia cells and KB human epidermoid carcinoma cells	0.72 and 0.42 µg/mL
Lissoclinamide 8 [55]	sea squirt *Lissoclinum patella*	Cytotoxicity against T24 (bladder carcinoma cells), MRC5CV1 (fibroblasts), and lymphocytes	6, 1, and 8 μg/mL
Mechercharmycin A [79]	marine bacterium *Thermoactinomyces* sp. YM3-251	Cytotoxic activity against A549 (human lung cancer) cells and Jurkat cells (human leukemia)	4.0 × 10^−8^ M and 4.6 × 10^−8^ M
Leucamide A [70]	marine sponge *Leucetta microraphis*	Cytotoxicity against HM02, HepG2, and Huh7 tumor cell lines	5.2, 5.9, and 5.1 μg/mL
Bistratamide H [50]	marine ascidian *Lissoclinum bistratum*	Cytotoxic activity against the human colon tumor (HCT-116) cell line	1.7 µg/mL
Patellamide E [58]	marine ascidian *Lissoclinum patella*	Cytotoxicity against human colon tumor cells in vitro**	125 µg/mL
Microcyclamide [35]	cultured cyanobacterium *Microcystis aeruginosa*	Cytotoxicity against P388 murine leukemia cells	1.2 µg/mL
Dolastatin E [76]	sea hare *Dolabella auricularia*	Cytotoxicity against HeLa-S_3_ cells	22–40 μg/mL
Aerucyclamide A [38]	freshwater cyanobacterium *Microcystis aeruginosa* PCC 7806	Antiparasite activity against *Plasmodium falciparum* K1 and *Trypanosoma brucei rhodesiense* STIB 900	5.0 and 56.3 μM
Aerucyclamide B [38]	freshwater cyanobacterium *Microcystis aeruginosa* PCC 7806	Antiparasite activity against *Plasmodium falciparum* K1 and *Trypanosoma brucei rhodesiense* STIB 900	0.7 and 15.9 μM
Aerucyclamide C [38]	freshwater cyanobacterium *Microcystis aeruginosa* PCC 7806	Antiparasite activity against *Plasmodium falciparum* K1 and *Trypanosoma brucei rhodesiense* STIB 900	2.3 and 9.2 μM
Aerucyclamide D [38]	freshwater cyanobacterium *Microcystis aeruginosa* PCC 7806	Antiparasite activity against *Plasmodium falciparum* K1 and *Trypanosoma brucei rhodesiense* STIB 900	6.3 and 50.1 μM
Aerucyclamide A, B and C [37,38]	freshwater cyanobacterium *Microcystis aeruginosa* PCC 7806	Grazer toxicity against the freshwater crustacean *Thamnocephalus platyurus*	30.5, 33.8, and 70.5 μM
Aerucyclamide B and C [38]	freshwater cyanobacterium *Microcystis aeruginosa* PCC 7806	Cytotoxic activity against Rat Myoblast L6 cells	120 and 106 μM
Urukthapelstatin A [78]	marine-derived bacterium *Mechercharimyces asporophorigenens* YM11-542	Cytotoxicity against A549 human lung cancer cells	12 nM
Mechercharmycin A [79]	marine-derived bacterium *Thermoactinomyces* sp.	Cytotoxicity against A549 human lung cancer cells and Jurkat cells	4.0 × 10^-8^ M and 4.6 × 10^-8^ M
Ulithiacyclamide [56,117]	marine tunicate *Lissoclinum patella*	Cytotoxic activity against L1210, MRC5CV1, T24, and CEM cell lines (continuous exposure)	0.35, 0.04, 0.10, and 0.01 μg/mL
Ulicyclamide [117]	marine tunicate *Lissoclinum patella*	Cytotoxic activity against L1210 murine leukemia cells	7.2 μg/mL
Patellamide A [117]	marine tunicate *Lissoclinum patella*	Cytotoxic activity against L1210 murine leukemia and human ALL cell line (CEM)	3.9 and 0.028 μg/mL
Patellamide B, C [117]	marine tunicate *Lissoclinum patella*	Cytotoxic activity against L1210 murine leukemia cells	2.0 and 3.2 μg/mL
Venturamide A [34]	marine cyanobacterium *Oscillatoria* sp.	Antiparasitic activity against *Plasmodium falciparum, Trypanasoma cruzi* Cytotoxicity against mammalian Vero cells and MCF-7 cancer cells	8.2 and 14.6 μM 86 and 13.1 μM
Venturamide B [34]	marine cyanobacterium *Oscillatoria* sp.	Antiparasitic activity against *Plasmodium falciparum, Trypanasoma cruzi* Cytotoxicity against mammalian Vero cells	5.2 and 15.8 μM 56 μM
Bistratamides A and B [60]	aplousobranch ascidian *Lissoclinum bistratum*	Cytotoxicity against MRC5CV1 fibroblasts and T24 bladder carcinoma cells	50 and 100 µg/mL
Bistratamide M [61]	marine ascidian *Lissoclinum bistratum*	Cytotoxicity against breast, colon, lung, and pancreas cell lines	18, 16, 9.1, and 9.8 μM
Balgacyclamide A [33]	freshwater cyanobacterium *Microcystis aeruguinosa* EAWAG 251	Antimalarial activity against *Plasmodium falciparum* K1	9 and 59 μM
Balgacyclamide B [33]	freshwater cyanobacterium *Microcystis aeruguinosa* EAWAG 251	Antiparasitic activity against *Trypanosoma brucei* *rhodesiense* STIB 900	8.2 and 51 μM

^a^ MIC—minimum inhibitory concentration, ^b^ CNS—central nervous system, ^c^ KB—ubiquitous KERATIN-forming tumor cell subline, ^d^ ED_100_—effective dose for 100% inhibition.

**Table 3 marinedrugs-18-00329-t003:** Issues associated with marine peptide drug development.

Sr. No.	Associated Issue
1.	Low bioavailability and short half-life due to instability of peptides in the body
2.	Formulation challenges and synthesis challenges including aggregation and solubility problems
3.	Difficulty optimizing peptide length to pharmacologically useful levels for receptor activation
4.	Expensive synthesis and manufacturing cost
5.	Difficulty in delivering expected purities and yields

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
