# Peer review of "Natural Bioactive Thiazole-Based Peptides from Marine Resources: Structural and Pharmacological Aspects"

_marinedrugs, 2020, doi:10.3390/md18060329_

Round 1

Reviewer 1 Report

The manuscript from Dahiya et al. reviews in a comprehensive way the structural and pharmacological aspects of thiazole-based peptides of marine origine. Besides a few typos (eg line 141 there is a "be" missing between "to" and "crucial",line 866 "inhibitioin" instead than "Inhibition") the article is written in a clear and comprehensible way. I appreciate the effort made by the authors to thoroughly  review such a wide class of compounds and I found a good idea to insert tables that help in tracing information. I think that this work can be of interest for scientists working in the field of natural and bioactive compounds, as well as for researchers interested in modified peptides. However, there are paragraphs that are heavy to read and the figures result poorly informative and not well connected with the text. Based on the considerations above, I would recommend the publication of the present review in Marine Drugs, while advising the authors to address the following points:

1) paragraph 1.1 is a list of cyanobacteria and other sources of the peptides making the subject of this review. The estensive references are very useful, but the list is difficult to read and does not add much information. Thus I would eliminate this paragraph and include the info in 2-3 senteces at the beginning or at the end of the subsequent paragraph 1.2 (for example: "Various natural sources of TBPs and other heterocyclic rings containing cyclopolypeptides include cyanobacteria,(refs) marine sponges (refs), ascidians (refs), sea slugs (refs) and higher plants (refs)." It would already be enough and for the interested readers details can be reported directly in the bibliography)

2) paragraph 2.1 starts with a list of names and the first sentence ends referring to fig 1 which is two pages later: it is extremely complicate to relate the listed peptides to the structures, both because one needs to scroll back and forth, and because not all the peptides are represented in the figure. I would suggest to divide fig 1 into several figures and to insert them within the text, near the place where the peptide(s) is(are) described. I also recommend to add more informative captions to the figures.

3) paragraphs 2.2 and 2.3 are easier to read than 2.1, but suffer of the same problem described above: I would advice, here too, to divide figures 2 and 3 in several figures inserted within the corresponding paragraphs.

4) the following paragraphs 2.4-2.6 and 3 are interesting and contain important information, but the figure is again poorly connected with the text. I would suggest to either separate it in several figures (as recommended in point 2 and 3) or to insert a description under each structure.

Author Response

15/06/2020

Response to Reviewer 1 comments

Typo on line 115 (earlier line 141): ‘be’ is inserted between ‘to’ and ‘crucial’.

Typo on line 987 (earlier line 866): "inhibitioin" is corrected as "Inhibition".

Paragraph 1.1 between lines 89-92 is shortened, as suggested. Names of individual organisms are removed and corresponding citations are combined together.

Paragraph 2.1 between lines 127-312: Original Figure 1 is divided into Figures 1-6 (lines 156-160, 171-176, 211-216, 225-230, 244-248, 263-267) and their captions are also improved.

Paragraph 2.2 between lines 312-396: Original Figure 2 is divided into Figures 7-9 (lines 331-342, 361-368, 385-395) and their captions are also improved.

Paragraph 2.3 between lines 396-396: Original Figure 3 is divided into Figures 10-14 (lines 420-425, 442-446, 460-465, 489-497, 514-517) and their captions are also improved.

Paragraphs 2.4-2.6 contain no figures.

Paragraph 3 between lines 703-841: Original Figure 3 is divided into Figures 15-19 (lines 716-721, 760-767, 785-791, 799-805, 822-828) and their captions are also improved.

DR. RAJIV DAHIYA

(Corresponding Author)

Reviewer 2 Report

I would like to congratulate the authors for undertaking this heavy task of providing an ensemble view of this class of active molecules. Very useful review especially for those looking to have a background check in this field. 

As a minus, I would mention the length of the document, but because it is well divided into subsections it is more accessible. However, maybe it would be useful  to have a small summary at the beginning. 

Also, some minor English mistakes are present, for example

Line 141 “was found to crucial”

Line 311 "othet”

Author Response

15/06/2020

Response to Reviewer 2 comments

A small summary of the study is included at the beginning (lines 74-87).

Typo on line 115 (earlier line 141): ‘was found to crucial’ is corrected as ‘was found to be crucial’.

Typo on line 320 (earlier line 311): ‘othet’ is corrected as ‘other’.

DR. RAJIV DAHIYA

(Corresponding Author)